# Impact of correcting sub-daily climate model biases for hydrological studies

Mina Faghih [1], François Brissette [1], Parham Sabeti [1]

[1] Hydrology, Climate and Climate Change Laboratory, École de technologie supérieure, 1100 Notre-Dame West st., Montreal (Canada) H3C1K3

Correspondence to: Mina Faghih (email: Mina.Faghih.1@ens.etsmtl.ca; Tel.: +15149949710)

**Abstract.** The study of climate change impact on water resources has accelerated worldwide over the past two decades. An important component of such studies is the bias correction step, which accounts for spatiotemporal biases present in climate model outputs over a reference period, and which allows realistic streamflow simulations using future climate scenarios. Most of the literature on bias correction focuses on daily scale climate model temporal resolution. However, a large amount of regional and global climate simulations are becoming increasingly available at the sub-daily time step, and even extend to the hourly scale, with convection-permitting models exploring sub-hourly time resolution. Recent studies have shown that the diurnal cycle of variables simulated by climate models is also biased, which raises issues respecting the necessity (or not) of correcting such biases prior to generating streamflows at the sub-daily time scale. This paper investigates the impact of bias-correcting the diurnal cycle of climate model outputs on the computation of streamflow over 133 small to large North American catchments. A standard hydrological modeling chain was set up using the temperature and precipitation outputs from a high spatial (0.11º) and temporal (1-hour) regional climate model large ensemble (ClimEx-LE). Two bias-corrected time series were generated using a multivariate quantile mapping method, with and without correction of the diurnal cycles of temperature and precipitation. The impact of this correction was evaluated on three small (<500 km$^2$), medium and large (>1000 km$^2$) surface area catchment size classes. Results show relatively small (3 to 5%) but systematic decreases in the relative error of most simulated flow quantiles when bias-correcting the diurnal cycle of precipitation and temperature. There was a clear relationship with catchment size, with improvements being most noticeable on the small catchments. The diurnal cycle correction allowed for hydrological simulations to accurately represent the diurnal cycle of summer streamflow in small catchments. Bias-correcting the diurnal cycle of precipitation and temperature is therefore recommended when conducting impact studies at the sub-daily time scale on small catchments.

**Keywords**: Hydrological modeling; Bias correction; Diurnal cycle; Impact study; ClimEx large ensemble

# 1 Introduction

The potential impacts of climate change have become a crucial concern for public safety, the environment and the economy of the twenty-first century (Raza et al., 2019; Walsh et al., 2019; Vogel et al., 2019). There is evidence that the hydrological cycle has already been significantly influenced by the changing climate in many regions, and it has become an important issue for water resource managers and policy makers (Yira et al., 2017; Zhao et al., 2019; Qiu et al., 2019). In particular, it is expected that the frequency of extreme precipitation and convective storms will increase at the local and regional scales, and particularly in mid to high latitudes (Martel et al., 2020; Pfahl et al., 2017; Myhre et al., 2019; Sarhadi and Soulis, 2017;

Barbero et al., 2017; Prein et al., 2017). Changes in extreme precipitation and patterns of convective storms will in turn impact flood risk (Quintero et al., 2018; Prein et al., 2017; Westra et al., 2014). To properly resolve extreme summer-fall convective

precipitation, a sub-daily modelling time step is required for most applications (Sunyer et al., 2017; Beranová et al., 2018; Bao et al., 2017). In hydrology, this is particularly true for small watersheds, which have a sub-daily response time, and are most likely to be affected by the anticipated sub-daily amplification of precipitation extremes (Yuan et al., 2019). In order to better adapt to the consequences of a changing climate, and to mitigate the future flood risk related to precipitation extremes on small watersheds, it is critical to consider a sub-daily time step for the entire hydro-climatic modeling chain. (Blenkinsop et al., 2018;

Beranová et al., 2018) ).

General circulation models (GCMs) and Earth System Models (ESMs) are invaluable tools for simulating the present and future climates (Panday et al., 2015; Alfieri et al., 2015). These models do however require substantial computational power and disk space, which significantly limits both the spatial and temporal resolution at which they can be run, and the frequency at which their outputs can be archived. This is particularly the case for GCMs and ESMs which are run at the global scale.

This explains why output data from these models have typically been limited to a relatively coarse spatial resolution of $1^{\circ}$ or more($\geq$100km), and been archived at the daily time scale. These spatial and temporal resolutions are too coarse to allow studying the potential hydrological impacts of climate change on small catchments (Trzaska and Schnarr, 2014; Bajracharya et al., 2018; Fatichi et al., 2014)

To overcome this issue, regional climate models (RCMs) have been used to dynamically downscale GCM outputs at a higher

spatial and temporal resolution over limited area domains. RCMs can better take into account local topography, land sea contrast, soil properties, and land cover, which impact surface forcing and physical processes. The spatial resolution of RCMs is generally in the range of 0.1 to $0.5^{\circ}$ (10 to 50km), with typical temporal resolutions of 3 to 6 hours, which are suitable for forcing hydrological models on relatively small catchments. More recently, the use of convection-permitting RCMs has bridged the resolution gap to $0.02^{\circ}$ (2 km) or below (Van Lipzig and Prein–Nicole, 2015; Chan et al., 2014; Kendon et al.,

2017). This increase in spatial resolution requires a corresponding increase in temporal resolution (for numerical stability), and such models are therefore limited to even smaller computational domains.

To properly assess climate model uncertainty, several multi-model (GCM and RCM) ensembles (e.g. CMIP5/6, CORDEX) have been used to address the uncertainty originating from greenhouse gas emission scenarios and structural climate model uncertainty. Internal climate variability is a third source of uncertainty, which can be studied with a multi-member ensemble

from a single climate model and single greenhouse emission scenario. Each member of the ensemble originates from micro and macro perturbations to initial conditions (Deser et al., 2012; Deser et al., 2020). Using multi-member ensembles has become increasingly popular in the analysis of the impact of internal variability, as well as for exploring the impact of extreme climate events such as extreme precipitation, since these ensembles provide many ergodic climate realizations from which to sample large numbers of extreme events (Zhao et al., 2020; Martel et al., 2020; Shen et al., 2018).

All global and regional climate model outputs are biased to some extent when compared to observations over a common reference horizon. These biases have a complex spatial and temporal structure (Chen et al., 2013b; Maraun, 2016; Ashfaq et

al., 2010; Wang et al., 2014). Therefore, a bias correction step is considered as a prerequisite for most climate change impact assessment studies. A wide range of bias correction techniques are available, extending from simple scaling methods to more advanced trend-preserving multivariate distribution mapping approaches. There is a significant body of literature on bias correction methods and several inter-comparison studies have been published (Fang et al., 2015; Lafon et al., 2013; Chen et al., 2013b; Maraun, 2016; Ajaaj et al., 2016; Bárdossy and Pegram, 2011). However, this is no longer true since climate model outputs are increasingly available at sub-daily time steps. A very limited number of studies has looked at bias correction of sub-daily climate model outputs, but the focus has been on correcting sub-daily annual maximum values (e.g. Li et al., 2017; Requena et al., 2021). Annual maximum values are important since they are used to determine the return period of extreme events for engineering design. For example, Li et al. (2017) showed that bias correcting the hourly annual maximum rainfall was recommended. It is well recognized that climate model biases are not constant in time, and as a result, different correction factors are typically computed for each month, or using a moving window across a calendar year. It is also known that high-resolution climate models are also biased in the reproduction of the diurnal cycle of many variables (Scaff et al., 2019; Bannister et al., 2019). As climate models slowly continue their steady march towards the sub-daily resolution, interesting research questions must be tackled. Should we bias-correct the diurnal cycles of climate model outputs? If so how? Do we have reliable reference datasets at the sub-daily time scale? Will this even influence the results of impact studies?

To provide an answer to these questions, this paper examines the impact of bias-correcting the diurnal cycle on the hydrology of several North American catchments. It also examines how the spatial scale influences the dynamic response of watersheds to extreme precipitation. In general, smaller watersheds are more sensitive to intense short-duration storms, whereas streamflows from larger catchments are somewhat smoothed by the flood wave propagation routing process. Therefore, in principle, an accurate representation of the diurnal cycle should be more critical for smaller catchments. To investigate this further, a wide range of catchment sizes has been selected.

This paper is structured into three main sections. The methodology provides an overview of the study area, describes all datasets (observations and climate model) and presents the bias correction method chosen to correct the diurnal cycle. Section 3 presents all results, and section 4 provides a discussion of the main results as well as concluding remarks.

## 2 Material and methods

### 2.1 Study Area

This study was conducted over the eastern United States in a rectangular region within the computational domain of the high-resolution regional climate model used (see section 2.2 below for additional details). As described below, 133 MOPEX catchments were selected based on the criteria of having observed hydrometric and meteorological data with less than 5% of

missing data over a common 24-year reference period. These catchments are dispersed across 4 climate zones of the Köppen climate classification. The impact of the catchment size is examined in this study by classifying catchments into three groups: less than 500 km$^2$, between 500 and 1000 km$^2$ and more than 1000 km$^2$. Catchments smaller than 500 km$^2$ should have a clear sub-daily hydrological time response as compared to the larger catchments. Figure 1 presents the centroid location and relative size of each catchment. Basic catchment characteristics are presented in Table 1.

Insert Figure 1 here

Insert Table 1 here

## 2.2 Datasets

All the results presented in this paper are available at the hourly time step. All observations cover the 24-year 1980-2003 period, which is defined as the reference dataset.

### 2.2.1 Observed Data

Hourly observed precipitation and streamflow data were derived from the Model Parameter Estimation Experiment (MOPEX) (Duan et al., 2006). MOPEX hourly precipitation is a catchment-averaged value from the closest weather stations. The MOPEX database does not however provide the hourly temperature. Rather than interpolating daily maximum and minimum values to the hourly scale, we took hourly temperature data directly from ERA5 reanalysis (Lindsay et al., 2014). At the catchment scale,Tarek et al. (2020a) showed that the ERA5 temperature is just as good as estimates derived from weather stations for hydrological modeling. The mean of all ERA5 grid points within each catchment was computed for every hour.

### 2.2.2 Climate Model Data

This project uses the ClimEx Large Ensemble (Leduc et al., 2019). The Climate Change and Hydrological EXtremes project (ClimEx) is a 50-member regional large ensemble computed using the 5[th] generation of the Canadian Regional Climate Model (CRCM5). CRCM5 was used to dynamically downscale the 50 members of the Canadian Earth System Model (v2) large ensemble (CanESM2-LE) (Arora et al., 2011) to a 0.11º (12 km) spatial resolution (Leduc et al., 2019; Martel et al., 2017) . The temporal resolution of archived ClimEx data is one hour for precipitation and three hours for most other variables. The ClimEx ensemble provides a sample of 7500 years, with each member covering the 1951-2100 period under the RCP 8.5 scenario. In this study, hourly precipitation and 3-hour temperature data were extracted for all grid points within each catchment over the Climex Northeastern-North-American (NNA) domain. The ClimEx temperature was first interpolated to the hourly time step for all grid points by using Piecewise Cubic Hermite Interpolating Polynomials (Fritsch, 1985; Barker and

Mcdougall, 2020). Precipitation and temperature were then averaged at the catchment scale to be consistent with the observed data over the reference period.

## 2.3 Bias Correction

The N-dimension multivariate bias correction (MBCn) by Cannon (2018) was selected in this study to correct biases of hourly precipitation and temperature. MBCn was chosen because it is arguably the most advanced quantile-based multivariate bias correction method available (Meyer et al., 2019; Chen et al., 2018; Su et al., 2020; Cannon et al., 2020). MBCn (Cannon, 2018) is a multivariate generalization of quantile mapping that conveys all aspects of the distribution of observation data to the corresponding distribution from a climate model. MBCn preserves the climate model projection trends for all quantiles, which is a highly desirable property for climate change impact studies (e.g. Maraun, 2016).

 All members of the ClimEx large ensemble were pooled together to compute the bias correction factors for both precipitation and temperature. The correction factors were then applied to all the members of the ClimEx ensemble. As discussed by Ayar et al. (2021) and Chen et al. (2019), doing so preserves the internal variability of the ensemble. This paper is not directly concerned with the study of internal variability, but using a large ensemble allows the accurate empirical computation of extreme events with very large return periods (Martel et al., 2020). Since climate model biases are not constant across the annual cycle, different correction factors were computed for each month of the year.

In observance of the main objective of the present study, the MBCn bias correction method was applied in two different ways:

1.   Standard Bias Correction (SBC): For each calendar month, a single set of quantile correction factors was applied to all hourly data. This approach assumes that all climate model biases are constant across the diurnal cycle. In this variant, for each month, there is one set of quantile correction factors and all hourly values are corrected using this set.

2.   Diurnal Bias Correction (DBC): This variant specifically recognizes that climate model biases are not constant throughout the diurnal cycle (e.g. daylight biases may differ from nighttime biases). Bias corrections were therefore computed for each hour, using a 3-hour moving window to pool all hourly values within a given month before using the MBCn algorithm. This was performed to smooth the diurnal cycle of observations, and therefore remove some of the sampling noise in the observed data. In this variant, for each month, there are 24 sets of quantile correction factors (one for each hour).

## 2.4 Hydrological model (GR4H)

A hydrological model is needed to take and transform precipitation and temperature data into streamflow values. In this study, hourly streamflows were simulated by the GR4H (modèle du Génie Rural à 4 paramètres Horaire) hydrological model. GR4H is an hourly rainfall-runoff model derived from its daily time step sibling, GR4J (Perrin et al., 2003). GR4H is a lumped

conceptual model with two storage reservoirs and four free parameters which define the production and routing functions, and which must be calibrated. GR4H was coupled with the CEMANEIGE degree-day snow model to simulate snowpack accumulation and depletion. CEMANEIGE is a two-parameter snow model developed by Valéry (2010). The combination of these two models, GR4J (GR4H) and CEMANEIGE, has shown good performance in different studies throughout the world (Riboust et al., 2019; Youssef et al., 2018; Raimonet et al., 2018). GR4h requires precipitation, temperature and potential evapotranspiration (Westra et al., 2014) as hourly inputs (Van Esse et al., 2013). The Oudin Ep formulation (Oudin et al., 2005) was used here. The combination of this Ep formula with the GR4J hydrological model has been used successfully in many hydrological studies (Arsenault et al., 2018; Troin et al., 2018)

The calibration of the hydrological model was performed automatically on all catchments using the Shuffled Complex Evolution (SCE-UA) algorithm (Duan et al., 1994), which has been shown to be highly efficient in a wide variety of problems (e.g. Huang et al., 2018; Muttil and Jayawardena, 2008; Arsenault et al., 2014). The Nash-Sutcliffe Efficiency (NSE) criterion was used as the calibration objective function. The NSE criterion has been used in many studies, and represents a normalized root mean square error. It compares the hydrological model efficiency to the mean flow as a reference predictor, as shown in the following equation:

$$NSE = 1 - \frac{\sum_{t=1}^{T}(Q_{Sim}^t - Q_{Obs}^t)^2}{\sum_{t=1}^{T}(Q_{Obs}^t - \overline{Q}_{Obs})^2}$$

where $Q_{Sim}^t$ and $Q_{Obs}^t$ are respectively the simulated and observed discharges at time t and $\overline{Q}_{Obs}$ is the mean of the observed discharge.

NSE values range from negative infinity up to 1. A value of 1 indicates a perfect agreement between modeled and observed data, while a 0 value indicates that the hydrological model's performance is no better than what is obtained from using the mean streamflow value as a predicting model. The hydrological model was calibrated over the entire 24-year period following the recommendations of Arsenault et al. (2018). They showed that using the entire observation record for the calibration of a hydrological model results in a more robust parameter set than using a shorter period followed by a validation step. The often used split sample calibration/validation strategy was therefore not implemented in this study.

**3 Results**

Figure 2 presents the NSE criterion values obtained for the calibration procedure described above for the 133 catchments. Overall, the model calibration is good, with a mean NSE value of 0.78 across all catchments. 94.6% of the catchments have an NSE value above 0.7 and 36.9%, a value above 0.8. The smallest NSE value is 0.61. These results show that the hydrological model does a good job at simulating the hourly streamflow on the selected catchments.

Insert Figure 2 here


Figure 3 presents the observed and ClimEx simulated temperature diurnal cycles of a selected catchment for all four seasons (left-hand side), as well as the results of both bias correction approaches (right-hand side). The 50 members of the ClimEx ensemble are presented as a shaded envelope, with the ensemble mean as a solid line. Throughout this paper, time refers to the

catchment local time. The left-hand side shows that ClimEx simulates a good temperature diurnal cycle, which is fairly close to the observed ones and for all seasons. Over this catchment, ClimEx runs a warm bias, especially for spring, summer and fall. The warm bias tends to be larger during the nighttime. All members of the ClimEx ensemble are very close to one another, with a difference of only about 1 degree between the coldest and warmest members. The diurnal cycle of temperature is hardly affected by internal climate variability.


Insert Figure 3 here

The right-hand side of Figure (3, B1 to B4) presents the performance of Cannon (2018) multivariate bias correction (MBC)

with diurnal cycle bias correction (DBC in green) and standard bias correction (SBC in blue). The pooling of all the ClimEx members to derive a unique set of bias correction factors preserves the signature of internal variability, as can be seen by the width of the blue and green envelopes as compared to those of the gray envelope of uncorrected ClimEx values (A1 to A4). With the standard bias correction (SBC), all hourly values are corrected using common correction factors for each month. The bias correction then reduces to a simple vertical scaling, which reduces the mean daily bias to zero. However, hourly biases

remain: these biases are negative from 06h00 to 14h00, and positive from 14h00 to midnight. For the green curves, using a 3-hour moving window results in a diurnal cycle that is smoother than the observed one. This was a methodological choice made in order to filter out variability in the observations, likely resulting from sampling errors. Without the smoothing window, the bias-corrected diurnal cycle would have matched those of observations exactly.

Figure 4 presents the observed and ClimEx simulated precipitation diurnal cycles for the same catchment. The layout of Figure

4 is the same as for the temperature (Figure 3). Compared to the temperature, the simulated internal variability of precipitation is much larger, as shown by the width of the gray envelope on the left-hand side. Internal variability is largest for winter and fall, and smallest during summer. Precipitation differences between members can reach up to 100%, depending on the season and hour, highlighting the key role of internal variability in driving precipitation variability. Over this catchment, ClimEx precipitation is positively biased in winter and spring and negatively biased over the summer. Overall, there are large

differences between observed and simulated precipitation, and these differences extend to the diurnal cycle. Summer is the only season where observations and ClimEx have a similar diurnal cycle despite a 3-4 hour lag between the peaks of both cycles. ClimEx presents a strong spring diurnal cycle, which is however, absent in the observations. Winter and fall do not show clear diurnal cycles in both the observations and ClimEx. The large differences between the observations and ClimEx

outputs testify to the need for bias correction prior to using climate model outputs in hydrological models (or other impact

models).

Insert Figure 4 here

Just as in Figure 3, the right-hand side of Figure 4 presents the performance of the multivariate bias correction (MBC) with diurnal cycle bias correction (DBC in green) and standard bias correction (SBC in blue).. Just as before, SBC (blue) simply scales precipitation to correct for the mean daily biases, with no impact on the shape of the modeled cycle. DBC (green), on the other hand, corrects the hourly distributions such that the bias-corrected diurnal cycle of ClimEx matches the observed one. Since precipitation correction is multiplicative, the internal variability envelope appears to be smaller in winter and spring

because ClimEx is positively biased for these seasons. The reverse is observed for the summer season, when ClimEx is negatively biased. The relative internal variability ( around the ensemble mean ) remains the same before and after correction. Overall, both bias correction methods do what they were designed for efficiently. The transformation of the gray envelopes into the green ones highlights the strength of these distribution mapping approaches. The fact that they can shape severely biased distributions into completely different ones also raises important questions about their use, as will be discussed later.

Now that the bias correction efficiency has been established, we can look at the hydrological modeling to see if the correction of the diurnal cycle has any impact on the hydrological simulations. To this end, raw and bias-corrected hourly precipitation and temperature time series were used to force the GR4H hydrological model to generate streamflow time series. Since the ClimEx ensemble was forced by a GCM (instead of reanalysis), it is not possible to directly compare the hourly simulated streamflow series with ClimEx meteorological data against those simulated using the observed meteorology. For this reason,

the first comparison will be based on the mean annual hydrograph. Figure 5 shows the mean annual hydrographs for four catchments of different sizes. It shows streamflow observations (redline), as well as streamflow simulations from the hydrological model, using precipitation and temperature from three different sources. They are the uncorrected ClimEx data (grey envelope) and bias corrected ClimEx data with and without accounting for the diurnal cycle biases (DBC, light green envelope with ensemble mean in dark green, and SBC, light blue envelope with the ensemble mean as a dotted dark blue line).

Results show that the multivariate bias correction of precipitation and temperature translates into accurate streamflow simulations. The ensemble mean tracks very well with the mean observed hydrographs contained within the ClimEx envelope of internal variability. Observations (red line) display a larger variability since they only contain 23 years of data, whereas the ensemble mean for both DBC and SBC comprise 1150 years (50 members times 23 years), and are therefore much smoother. The internal variability envelopes for DBC and SBC are very close to one another, with the blue envelope almost perfectly

overlapping the green one. There are, however, small differences between the ensemble mean curves, indicating that taking the diurnal cycle biases into account impacts streamflow simulations to some extent. The largest differences are observed for the smallest catchment (upper right).

Insert Figure 5 here


The impact of diurnal cycle bias correction as a function of catchment size is illustrated in Figure 6, which shows typical results for a small (66 km$^2$) and large (3817 km$^2$) catchments. Those two catchments have been chosen are they differ mostly with respect to their size. They are located close to one another (Figure 6) and share common physiographical properties.

Insert Figure 6 here

The figure presents a one-month (July) snapshot of streamflow hydrographs for the mean member of the ClimEx ensemble, with standard and diurnal cycle bias correction. The upper graph shows the quicker reactivity of the smaller catchment to
meteorological inputs as compared to the larger one. More importantly, Figure 6 shows that the diurnal cycle correction has a larger impact on the smaller catchment when compared to the larger one. On larger catchments, the flow routing process acts as a low-pass filter, resulting in somewhat smoothed hydrographs, and blurring the difference between the two bias correction approaches. Figure 6, however, only shows that the diurnal cycle bias correction has an impact on streamflows, and not if this impact is beneficial. To figure out if the impact is beneficial, it is necessary to look at streamflow indicators.
Figure 7 presents the impact of correcting the diurnal cycle on the relative bias $B$ of mean annual simulated streamflow, as expressed by equation 1a and 1b:

$$B_{DBC} = \frac{\overline{Q_{climexDBC}} - \overline{Q_{obs}}}{\overline{Q_{obs}}} \times 100\% \qquad 1a$$

$$B_{SBC} = \frac{\overline{Q_{climexSBC}} - \overline{Q_{obs}}}{\overline{Q_{obs}}} \times 100\% \qquad 1b$$

In the above equations, $\overline{Q_{obs}}$ is the mean annual streamflow resulting from running the hydrological model with observed
precipitation and temperature, whereas $\overline{Q_{climexDBC}}$ and $\overline{Q_{climexSBC}}$ respectively represent the mean annual simulated streamflow using bias corrected ClimEx precipitation and temperature, with and without correcting the diurnal cycles of both variables. Figure 7 shows boxplots of the relative bias of mean annual streamflow, with and without (DBC and SBC) mean diurnal cycle correction, for the three catchment size categories. Results are not shown for the streamflow simulations without bias-corrections since the errors are up to two order of magnitudes larger than for the bias-corrected simulations. Each boxplot
represents the distribution of mean relative streamflow bias for the 133 catchments. The central box displays the 25[th], 50[th] (median) and 75[th] quantiles of the distribution, whereas the lower and upper whiskers show the 5[th] and 95[th] quantiles. Values below and above the 5[th] and 95[th] quantiles are shown as red circles and mean of the distributions are shown by purple crosses. Overall, the relative biases are relatively small across the board, indicating that the bias correction method does a good job at preserving the main characteristics of observed precipitation and temperature, at least in terms of hydrological modeling.

Results show that accounting for diurnal cycles biases has an important impact on the representation of the mean annual streamflow. Correcting the diurnal cycle lowers the relative bias and diminishes the spread of the bias estimates. Relative biases are mostly positive with standard bias correction, and tend to be slightly negative with the diurnal bias correction. The impact is particularly clear for the small and medium catchments. For the large catchments, the absolute value of the median bias remains similar (goes from positive to negative), but the spread is lower when correcting the diurnal cycle. This is

particularly clear for the central box ($25^{th}$ to $75^{th}$ quantiles). As shown in Figure 3, the climate model diurnal cycle of temperatures is flatter than for observations. Bias correcting the diurnal cycle results in higher mean daily temperature leading to increased evapotranspiration and decreased streamflow values, likely explaining the observed results.

Insert Figure 7 here


To further understand the impact of the diurnal cycle correction, Figure 8 shows similar results for low-flow and high-flow metrics. Low flows are represented by the $5^{th}$ and $10^{th}$ quantiles of the annual streamflow distribution for each catchment, and high flows, by the $95^{th}$ and $99^{th}$ quantiles. All four graphs of Figure 8 are in the same format as those in Figure 7. The results are therefore expressed as relative biases, and each boxplot represents the distribution of relative biases across all 133

catchments.
Low flows (upper row) are generally not well represented, with relatively large negative biases (mostly in the -10 to -30% range). The negative biases are larger for the smaller catchments. Correcting the diurnal cycle slightly increases the negative biases for the small and medium size catchments, but has a positive impact on spread across all catchments. This is once again particularly clear for the interquartile range. High flows (lower row) are much better simulated, with biases below 10% in most

cases, with the exception of Q99 for the small catchments, where the biases are predominantly positive and much larger (+10 to +30%). Correcting the diurnal cycle provides relatively small, but consistent, bias reduction, as well as a reduction of the spread for the medium and large size catchments.

Insert Figure 8 here


Finally, Figure 9 presents similar results for the 20-year return period flood. The $95^{th}$, $99^{th}$ and 20-year return period are all high-flow indicators. However, the first two represent relatively frequent high flow thresholds, with several days per year exceeding these values (18 and 3 days per year on average), whereas the 20-year return period threshold is an extreme value

threshold that is exceeded once every 20 years on average. The 20-year return period was evaluated with a Log-Pearson III distribution following USGS guidelines (Flynn et al., 2006). It was calculated from the simulated flows using observed precipitation and temperature as well as bias-corrected ClimEx outputs. Figure 9 shows that bias-corrected data do a good job preserving the signature of meteorological data leading to extreme events. The relative biases are small for the medium and

large size catchments, and slightly positive and a bit larger over the small catchments. Correcting the diurnal cycle provides
relatively small but systematic bias reduction across-catchment spread improvements. These improvements are larger for the
smaller size catchments.

Insert Figure 9 here


**4 Discussion**

The preceding section has presented a hydrological modeling comparison of the impact of bias-correcting (or not) the diurnal
cycle of precipitation and temperature modeled by a high-resolution regional climate model. Figures 3 and 4 show that bias
correction methods can correct deficiencies in the representation of the diurnal cycles of temperature- and precipitation-
modeled data. In the case of the temperature, ClimEx simulates a diurnal cycle with an amplitude similar to that of observations,
but with a clear bias and timing offset. Both are effectively corrected using the MBCn method. The case of precipitation is
more complicated as there are large differences between observations and modeled data. The MBCn method, by construct,
was able to perfectly map the climate model biased diurnal cycle onto the observed one. Considering the large differences
between both cycles, a valid question is whether or not this bias correction step should even be done. The differences observed
between both cycles are rooted in three possible causes: observation errors affecting the observed diurnal cycle, structural
errors in the modeling of precipitation in the climate model, and internal climate variability. Measuring precipitation is difficult
(Yang et al., 1999; Angulo-Martínez et al., 2018), and particularly so at the sub-daily scale. Measuring issues related to the
use of tipping bucket rain gauges have been reviewed by Segovia-Cardozo et al. (2021). Those issues are an underestimation
of total amounts, and especially so for high intensity rainfall and light drizzle, losses from evaporation and non-linear response
to rainfall intensity. In addition, at the sub-daily scale, the above may cause small shifts in the actual recording of small
precipitation. Hourly recorded data is not available at all weather stations, and when it is, records often typically suffer from
large amounts of missing data. Performing a reliable estimation of the diurnal cycle is therefore by no means a simple task. In
this work, we used catchment-averaged hourly precipitation from the MOPEX database. Catchment selection for inclusion
into the Mopex database was based on several quality control requirements, including quality precipitation data and minimum
station density. While we can assume that the quality of precipitation data is good (or at least better than average), we have no
way to quantitatively assess the quality of the observed diurnal cycle over the reference period. This also limits our ability to
evaluate the diurnal cycle from the climate model. Differences are however large enough to suspect potential problems in the
physical representation of precipitation in ClimEx. The GCM and RCM climate model structures do not include all
mechanisms leading to precipitation in the real world, and this may lead to large errors (Legates, 2014). Even at the 0.11°
resolution of ClimEx, convection has to be parameterized, potentially leading to significant errors in the representation of

larger precipitation quantiles. Knist et al. (2020) and Prein et al. (2016) showed that resolving convection in climate models led to a better representation of precipitation intensity and of the diurnal cycle of precipitation, for example. Maraun et al. (2017) make a compelling argument with respect to the selection/disqualification of climate models based on their ability (inability) to represent key physical processes leading to any variable under consideration. Bias-correcting unrealistically

simulated variables raises many important issues. Nevertheless, such issues are rather peripheral to the stated goal of this paper, which is to explore the impact of correcting (or not) the diurnal cycle of precipitation. The third factor explaining differences between observed and simulated precipitation cycles is the role of internal variability. Figure 4 shows that internal variability plays a very significant role in the representation of the diurnal cycle of precipitation. For the fall period, the difference between the observed and modeled cycles is smaller than the internal variability for most of the cycle. The large internal variability of

precipitation has long been recognized in many studies (Deser et al., 2012; Dai and Bloecker, 2019), and it shows that 30 years (23 in the case of this study) of observations may simply not be a long enough period to adequately represent the diurnal cycle of precipitation.

After bias correction, climate model precipitation and temperature outputs were used in a hydrological model to generate streamflows. Hydrological modeling results point to a relatively modest but consistent increase in hydrological modeling

performance for all metrics (with the exception of low flows) when the diurnal cycle of precipitation and temperature is corrected. The performance increase was clearly larger for the small catchments, but improvements were also seen for the medium and large size classes. The reasons for this improvement are not easy to pinpoint. Correcting the temperature diurnal cycle ensures a more realistic representation of the daily cycle of evapotranspiration, which may explain the better representation of the mean annual streamflow discharge. We can gain some insights by looking at the diurnal cycle of

streamflow for summer (JJA) for one small and one large catchment, as shown in Figure 10. Small catchments are known to have such a cycle, where increased evapotranspiration in the afternoon (resulting from the strong temperature diurnal cycle) leads to a corresponding reduction of streamflow. It can be seen that the streamflow cycle is very well modeled for the small catchment when the diurnal cycle of both variables is corrected. For the large-size catchment, the diurnal streamflow cycle is flat for both observed and simulated streamflow. This shows that the catchment response time (flow routing transfer time or

time of concentration) is too large for the day-time increased evaporation to show at the basin outlet. The small differences induced by the diurnal cycle of precipitation and temperature data are smoothed out during flow routing to the basin outlet. The internal variability of precipitation is transferred to streamflow, as represented by the large envelope from the 50 members of the ClimEx ensemble.

Insert Figure 10 here

The absence of performance improvements for the low flow criterion can be partly explained by methodological choices. Modeling low flows is a more difficult task than modeling high flows, especially for conceptual models whose simplified structure is ill-suited to accurately represent the contribution of groundwater, which is complex, heterogeneous and sometimes

dominant in the absence of precipitation. It is also well-known that the NSE criterion that was chosen for the hydrological model calibration is more sensitive to high-flows (Krause et al., 2005; Muleta, 2012). Since modeled low flows displayed large biases with and without bias correction of the diurnal cycle, we do not believe that discussing badly modeled streamflow metrics is very relevant. A discussion on low flows would be better served by using a hydrological model targeted at droughts, either with a different model structure or using a different objective function during calibration.

There are many limitations to this study. A single climate model was used and our results should be replicated with other climate models. Potential differences may be related to bias correction and hydrological modeling. No bias correction method can correct all statistics and particularly so when it comes to joint distribution properties (P and T in this case). In addition, hydrological models are good spatial integrators, but they are sensitive non-linear integrators. As such, small changes between two climate models (e.g. spatial resolution, interannual variability) could ultimately results in different streamflow simulations.

While dramatically different results using other climate models are not expected, a different sensitivity to catchment size could possibly be observed. On the other hand, there are still not many climate model runs available with a high enough temporal and spatial resolution to apply to the study of small catchments, where the amplification of extreme precipitation is more likely to become critical as the climate becomes warmer. There are even fewer large ensembles being run at those fine resolutions. As shown in this paper, using a large ensemble shines a bright light on the role of internal climate variability in defining an

accurate diurnal cycle for precipitation. The importance of internal variability and how it brings irreducible uncertainty to the bias correction process has been discussed in details by (Chen et al., 2016; Chen et al., 2015; Maraun, 2012; Teutschbein and Seibert, 2013; Chen et al., 2018). A single bias correction method was used in this study. It is well-known that the choice of a bias correction method has implications, which are often very significant, on streamflow metrics, and that a large amount of uncertainty can arise from this choice (Chen et al., 2013a; Iizumi et al., 2017). For small catchments, we believe that using a

multi-variate method is highly desirable as preserving correlations between precipitation and temperature is key for an adequate representation of the diurnal cycle of key variables such as streamflows (as shown in Figure 10, for example). Small catchments modeled at the sub-daily scale would be very good targets to allow testing the advantage of multi-variate bias correction methods against univariate ones. Considering the subtle non-linear interactions between precipitation and temperature when modeling streamflows, it is possible that the improvements shown here in the representation of streamflows on small

catchments may not have been realized using a univariate correction. This is something which could be tested in future work. Hourly temperature from the ERA5 reanalysis was used instead of observations from stations. However, at the catchment scale, recent work at the daily temporal scale (Tarek et al., 2020a, b) showed that the ERA5 temperature was as good as, or better than, temperature gridded datasets derived purely from weather station observations. In addition, Lompar et al. (2019) showed that using the ERA5 hourly temperature to replace missing data in observed time series led to very low RMSE values.

This good performance of hourly temperature data is not entirely surprising considering that the surface temperature is assimilated by ERA5 and that the surface temperature can relatively easily be inferred from geopotential heights, which are typically well reproduced by reanalysis.

One important remaining limitation of this work lies in the bias-correction not having been evaluated in a split-sample methodology. The efficiency of any bias correction scheme on an independent period depends on the stationarity of the biases.

It has been shown in many studies that climate model biases are not constant in time (e.g.Wan et al., 2021; Maraun, 2012) and that non-stationarity can be amplified when using a hydrological model to simulate streamflows (Hui et al., 2020). The results presented here show that bias-correcting the diurnal cycle results in streamflow simulation improvements when tested on a common time-window with that of the bias correction process. Performing the same test on a different time window may impact the bias correction of the diurnal cycle of precipitation and temperature. In particular, the diurnal cycle of precipitation

is not-stationary due to internal variability (as shown in Figure 4), and it is possible that the advantages of the sub-daily bias correction method may be somewhat reduced when tested over an independent validation period, as found by Chen et al., (2018) in a comparison study of multivariate vs univariate bias correction methods. On the other hand, the diurnal cycle of temperature, which controls evapotranspiration (an important part of the diurnal streamflow cycle) is much less affected by internal variability (Figure 3).

In light of the above results, and despite the limitations of this study, some recommendations can be made to climate change impact modelers concerned with the impact of extreme precipitation on small catchments. For catchments smaller than 500 km$^2$, a sub-daily hydrological modeling step is generally required for a good simulation of the flood peak and timing. For such catchments, the bias correction should include a step to account for differences between the observed and modeled diurnal cycles of temperature and, to a lesser extent, precipitation. Climate models do generate (as shown here) a realistic temperature

diurnal cycle, and correcting for differences in timing and magnitude will ensure that the daily cycle of potential evapotranspiration matches that of observations. As discussed above, bias-correcting the diurnal cycle of precipitation is a bit more controversial. Taking into account the large internal variability of precipitation, as well as the potential issues surrounding the reliability of modeled precipitation, and especially extreme precipitation under a parameterized deep convection, arguments could be advanced from either side. Considering that these problematic issues also exist at the daily scale, and that bias

correction of precipitation at this time scale is almost universally performed in impact studies, we feel that bias-correcting the diurnal cycle of precipitation is likely the best recommendation. A comparison between correcting only the temperature diurnal cycle versus correcting both the precipitation and temperature could help in figuring out the variable from which most of the improvement is derived.

The issue of climate model resolution also needs to be raised. Climate model resolution has been steadily improving and there

is hope that with a higher resolution, the need for bias correction will be lessened ((Lucas-Picher et al., 2021). There are however computational physical limits as to how rapidly model resolution can decrease. Model resolution also competes with added model complexity, leading to a convergence between GCMs and ESMs (Bierkens, 2015) at the global modelling scale, rather than a sharp decrease in resolution. Regional climate models have seen the largest increase in spatial resolution, albeit at the expense of a progressively smaller computational domain. Climate model improvements have been shown to reduce

biases. These improvements come from the increased resolution (e.g. Lucas-Picher et al., 2017) resulting in a better representation of local topography and land surface, and from better physics (e.g. Kendon et al., 2017). However, climate

models remain an imperfect representation of the real climate system, and the sensitivity of impact models (e.g. hydrological model) to input data (e.g. precipitation, temperature) will still require some level of post-processing to insure realistic outputs from impact models. The ClimEx ensemble used in this study comes from a high-resolution regional climate model and quite

clearly requires bias correction, showing that spatial resolution is not the only piece of the puzzle. Using uncorrected ClimEx data results in unrealistic streamflow simulations (e.g. Figure 5). However, with better and higher-resolution models, there is hope that post-processing methods will only end up correcting minor model deficiencies, and not correcting bad physics over a given area (e.g. Maraun et al., 2017) such as an incorrectly modeled precipitation annual cycle for example. Increasing spatial resolution has however opened the door to convection-permitting models, which require a resolution of around 0.03° (3-4km)

or better to resolve convection without the need for parametrization. Convection-permitting models are becoming more common and have shown to improve the representation of precipitation and extreme precipitation ((Lucas-Picher et al., 2021). With the better physics of these models, it is likely that bias-correcting the daily cycle of precipitation will still be needed, but will be done for the right reasons, rather than to correct for sometimes implausible large biases. For larger catchments (> 500 km$^2$), results have shown that improvements linked to the diurnal cycle correction become progressively smaller. For sub-daily

hydrological modeling, it is however recommended to correct the diurnal cycle of temperature to ensure adequate representation of the potential evapotranspiration diurnal cycle. Correcting the daily cycle of precipitation is unlikely to make a big difference on streamflow metrics, considering the smoothing impact of flow routing. However, no ill effect of the diurnal cycle correction was observed for the medium to large catchments in this study. For those catchments, even though it was not investigated, it is likely that the relatively small improvements noted originated from the correction of the temperature daily

cycle and not from precipitation.

## 5 Conclusion

This paper investigated the impact of bias-correcting the diurnal cycle of a climate model on the computation of streamflow over 133 small to large catchments, using a high spatial (0.11°) and temporal (1-hour) regional climate simulation (ClimEx-

LE) over Eastern North America. The ClimEx regional climate model simulated a very realistic temperature diurnal cycle, but with timing and amplitude biases. There were however large differences between the simulated and observed diurnal cycles of precipitation. These differences result from a combination of observation errors, internal variability of precipitation and an inadequate representation of physical processes leading to precipitation by the climate model. These biases were successfully corrected using a multivariate quantile mapping method. The impact of bias-correcting (or not) the diurnal cycle of

precipitation and temperature was evaluated on small (<500 km$^2$), medium and large (>1000 km$^2$) catchments. Results indicate that correcting the diurnal cycle results in better streamflow simulation, especially for smaller catchments, which have a definite sub-daily response time. For the small catchments, the relative error between observed and simulated flow quantiles was reduced. For example, the median reduction was 5% for the 95[th] and 99[th] quantiles, and 4% for the median value of the

20-year flood across all small catchments. For larger catchments, bias-correcting the diurnal cycle only results in minor streamflow improvements. Despite the large differences in the diurnal cycles of observed and simulated precipitation, and the limitations of climate models in generating precipitation with parameterized convection, we nonetheless recommend bias-correcting the diurnal cycle of both temperature and precipitation when conducting climate change impact studies on small catchments at the sub-daily time step.

# 6 Appendix

Insert Appendix 1 here

# 7 Code and data availability

The MOPEX climate and streamflow database can be downloaded from the following link: (https://hydrology.nws.noaa.gov/pub/gcip/mopex/US_Data/ ) (Duan et al., 2006)

ERA5 data are available on the Copernicus Climate Change Service (C3S) Climate Data Store: https://cds.climate.copernicus.eu/cdsapp#!/dataset/reanalysis-era5-single-levels?tab=form (Hersbach and Dee, 2016).

ClimEx data can be downloaded from: https://www.climex-project.org/en/data-access

The GR4J model (Perrin et al., 2003) and CemaNeige snow module (Valéry et al., 2014) are available on the Matlab File Exchange:https://www.mathworks.com/matlabcentral/fileexchange/61720-gr4j-rainfall-runoff-model-deterministic-and-stochastic-methods-with-matlab.

The SCE-UA global optimization algorithm can be downloaded from:

https://www.mathworks.com/matlabcentral/fileexchange/7671-shuffled-complex-evolution-sce-ua-method

# 8 Acknowledgements

This work was partly financed through the ClimEx project funded by the Bavarian State Ministry for the Environment and Consumer Protection. The authors acknowledge the contributions from the Canadian Centre for Climate Modelling and Analysis [Environment and Climate Change Canada (ECCC)] for simulating and making available the CanESM2-LE used in this study, and the Canadian Sea Ice and Snow Evolution Network for proposing the simulations. The authors would also like

to thank the Ouranos Consortium for helping with data transfers. The CanESM2-LE dataset is now available on the ECCC website (http://crd-data-donnees-rdc.ec.gc.ca/CCCMA/products/CanSISE/output/CCCma/CanESM2/). The CRCM5 was developed by the ESCER Centre at Université du Québec à Montréal (UQAM; www.escer.uqam.ca) in collaboration with
ECCC. Computations with the CRCM5 for the ClimEx project were made on the SuperMUC supercomputer at the Leibniz Supercomputing Centre (LRZ) of the Bavarian Academy of Sciences and Humanities. The operation of this supercomputer is funded via the Gauss Centre for Supercomputing (GCS) by the German Federal Ministry of Education and Research and the Bavarian State Ministry of Education, Science and the Arts.

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

**10 Figures and table**

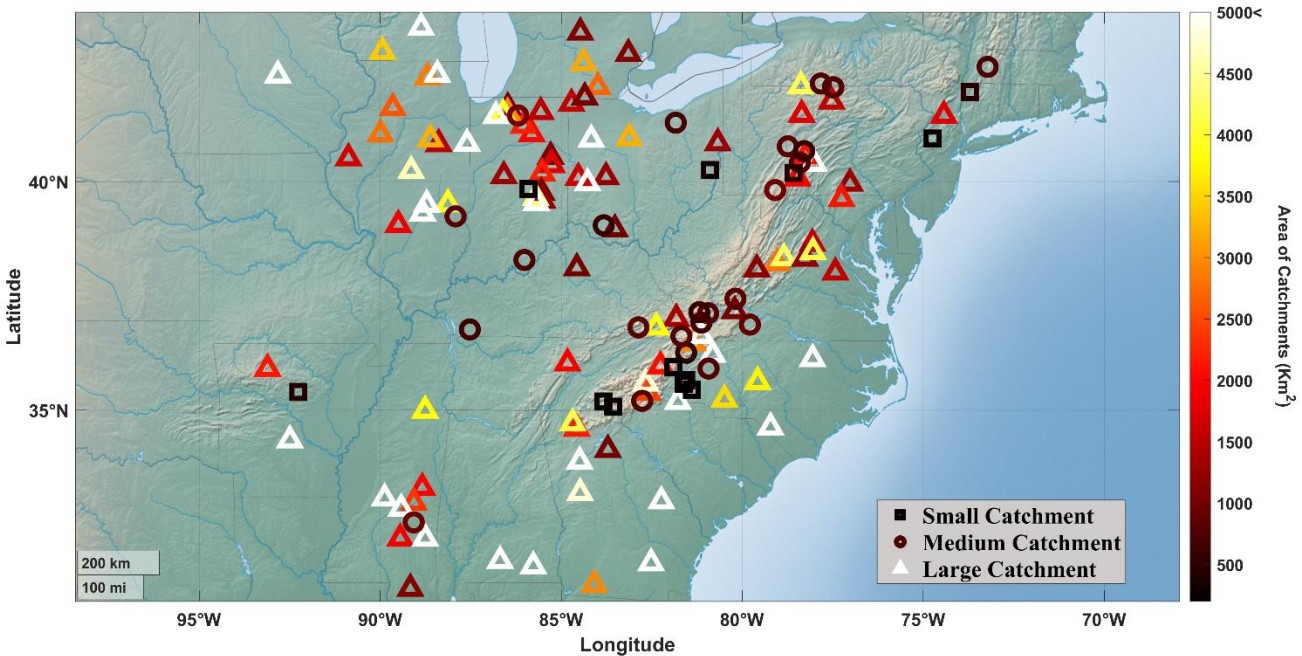

**Figure 1: Distribution of catchments across North Eastern America. Squares, Circles and triangles symbols correspond to small,**
**medium and large catchments respectively.**

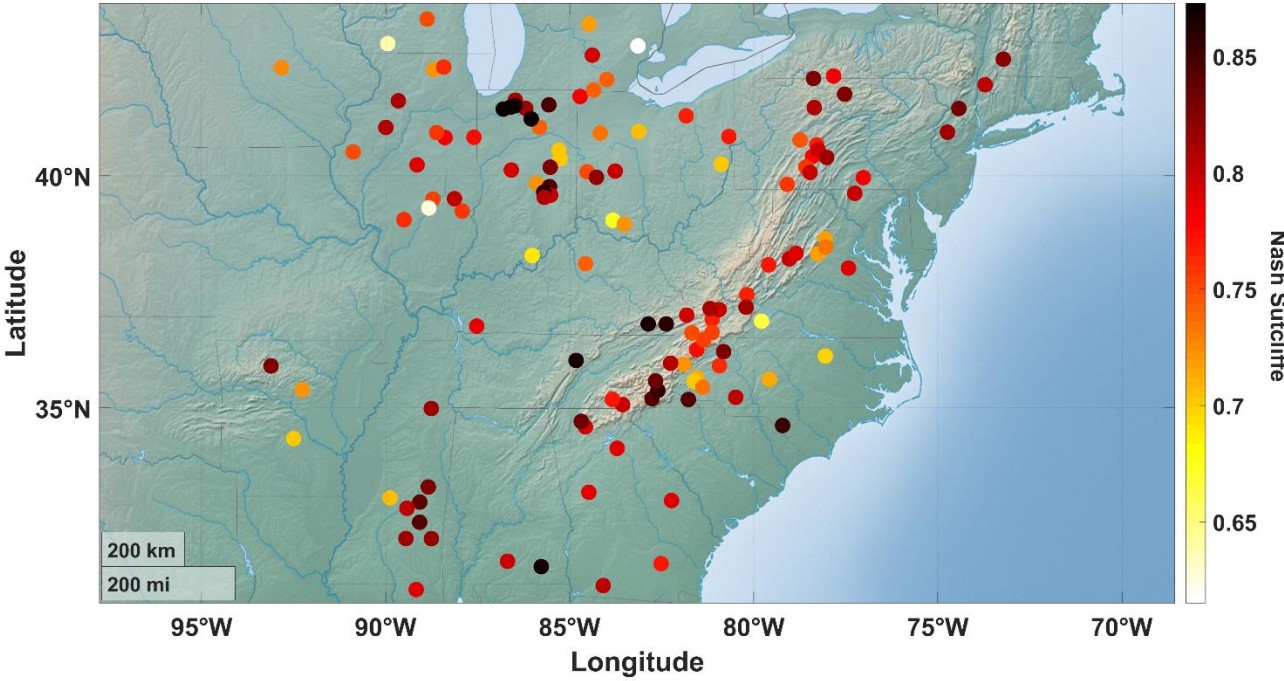

820

**Figure 2: NSE calibration results for all catchments.**

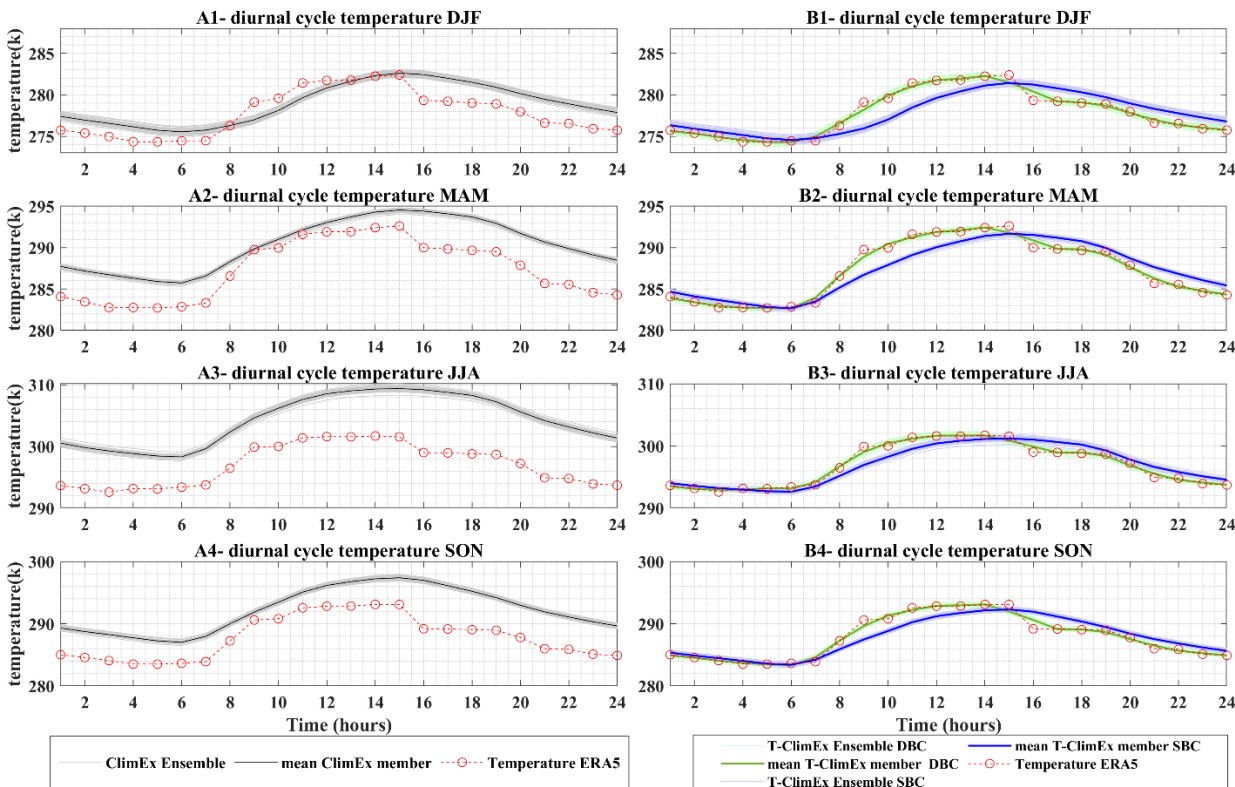

**Figure 3: Annual diurnal cycle of temperature before bias correction (first column: A1 to A4) and after bias correction (second column: B1 to B4) for catchment 02143040. Each row corresponds to a different season: DJF (December, January, February), MAM (March, April, May), JJA (Jun, July, August), SON (September, October, November). The right hand side shows both bias correction methods: Standard Bias Correction (SBC) and Diurnal Bias Correction (DBC). The observations (ERA5) are shown in red. Raw (uncorrected) ClimEx data is in grey, SBC is in blue and DBC is in green. The envelope defined by all 50 ClimEx members are shown in the corresponding light colours, whereas the dark coloured lines display the ensemble mean. Time is local with 24h corresponding to midnight.**

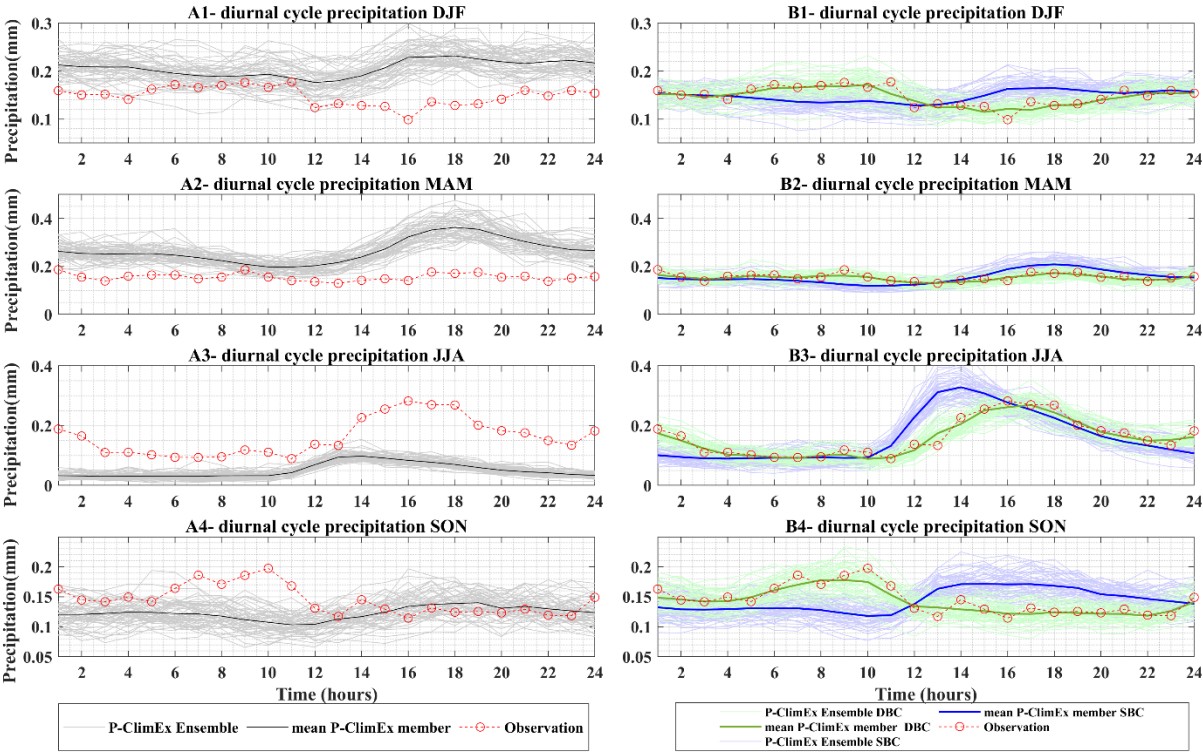

**Figure 4: Annual diurnal cycle of precipitation before bias correction (first column: A1 to A4) and after bias correction (second column: B1 to B4) for catchment 02143040. Each row corresponds to a different season: DJF (December, January, February), MAM (March, April, May), JJA (Jun, July, August), SON (September, October, November). The right-hand side shows both bias correction methods: Standard Bias Correction (SBC) and Diurnal Bias Correction (DBC). The observations are shown in red. Raw (uncorrected) ClimEx data is in grey, SBC is in blue and DBC is in green. The envelope defined by all 50 ClimEx members are shown in the corresponding light colours, whereas the dark coloured lines display the ensemble mean. Time is local with 24h corresponding to midnight.**



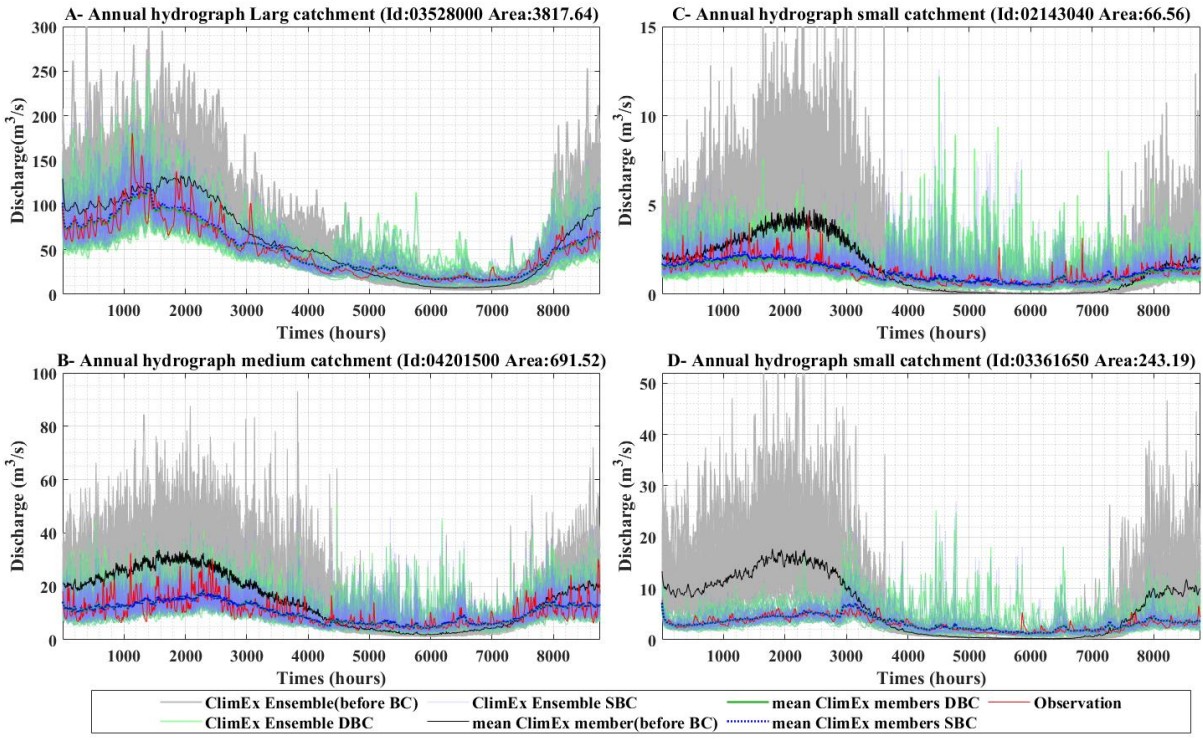

**Figure 5: Hydrograph annual cycles for four selected catchments. Catchments A and B are classified as large and medium size respectively. Catchments C and D are classified as small. 0 represents January first at 0h00, and 8760 is December 31st at 24h00.**

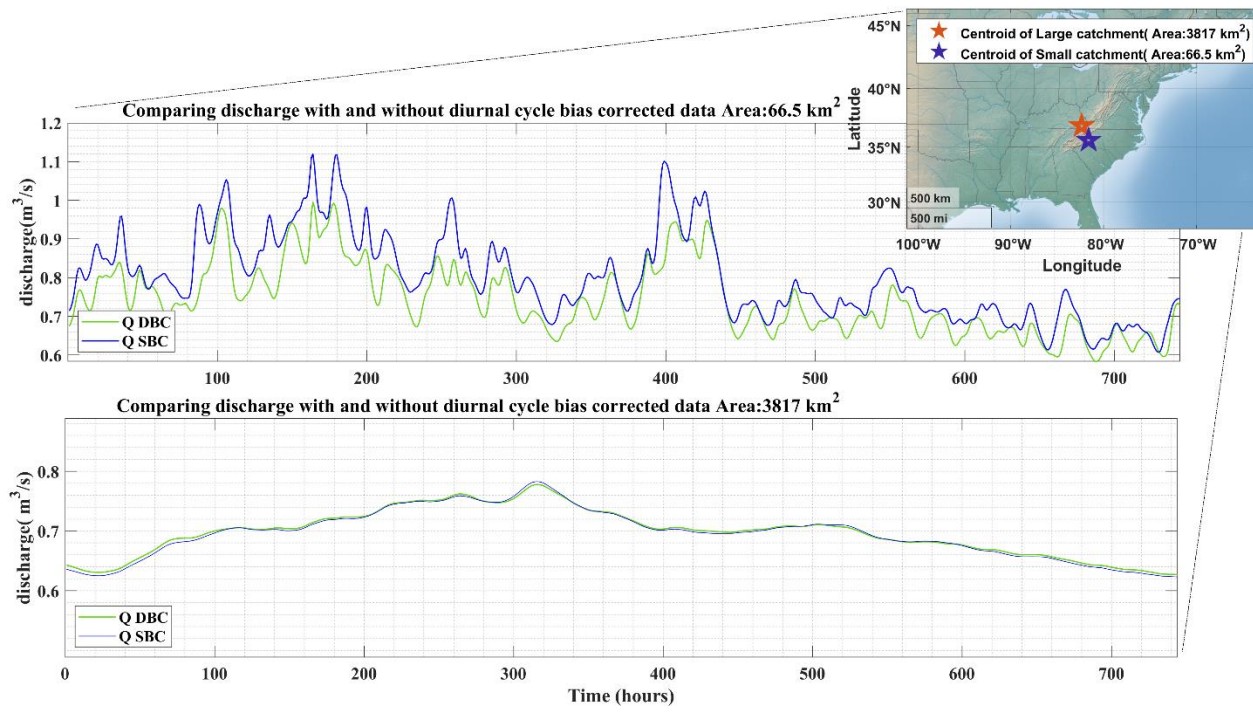


**Figure 6: Hydrographs of two sampled catchments (small and large size surface area) for the month of July (744 hours = 31 days × 24 hours).**

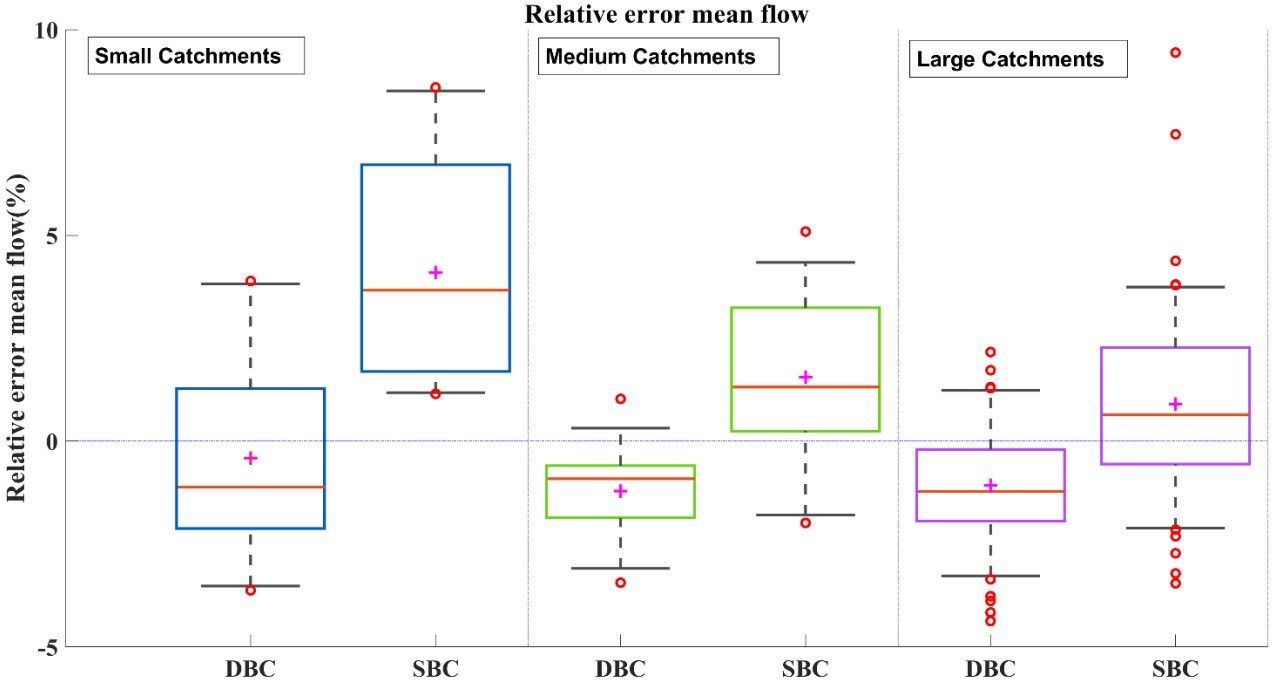

Figure 7: Comparing relative error of mean flow with diurnal cycle bias correction (DBC) and standard bias correction (SBC) in three area categories.

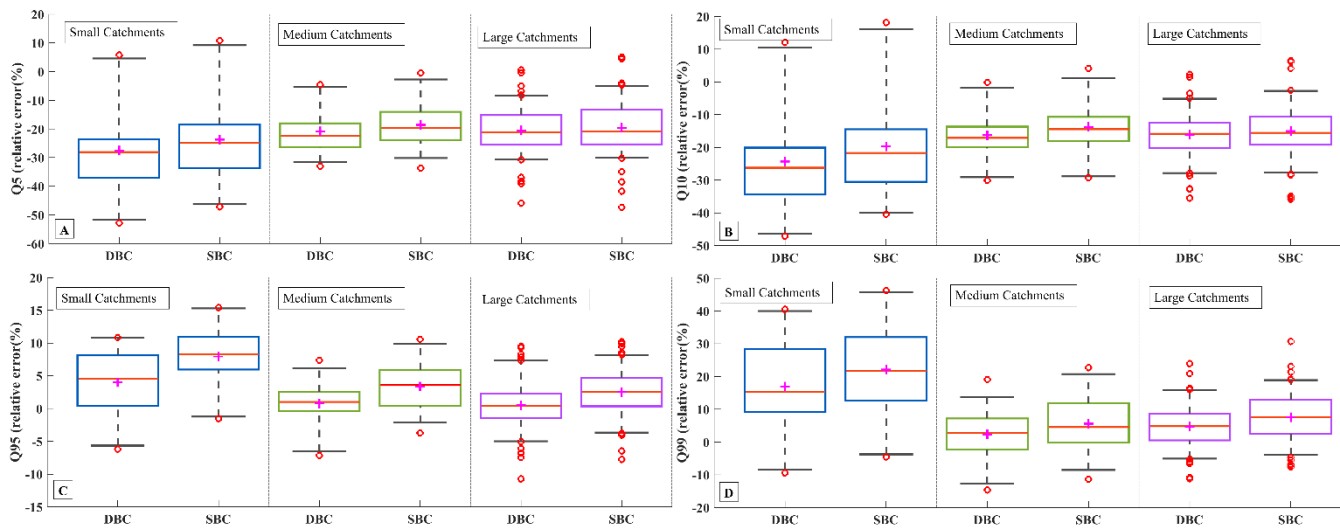

**Figure 8: Distribution of the relative error ((model-obs)/obs × 100%) corresponding to flow quantiles Q5 (A), Q10 (B), Q95(C) and Q99(D). Boxplots for both bias correction methods (DBC and SBC) are constructed from the distribution of relative errors from all catchments within each size class (small, medium, and large).**

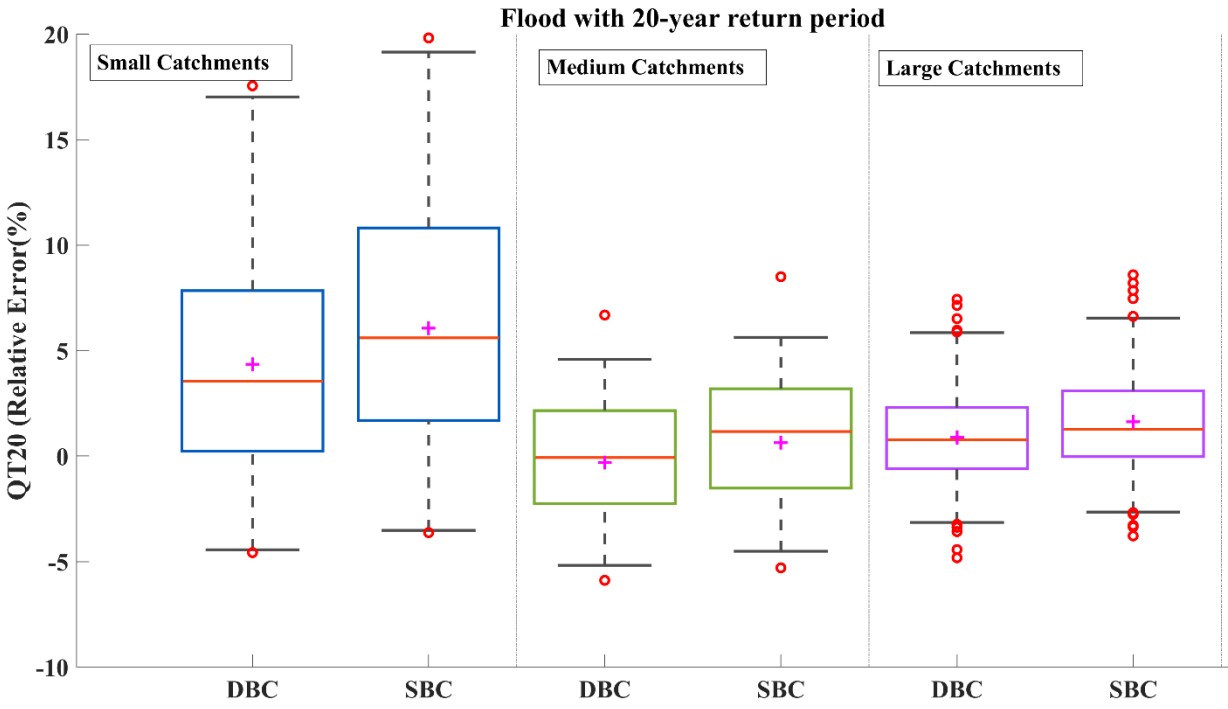

**Figure 9: Distribution of the relative error ((model-obs)/obs × 100%) for the 20-year flood QT20. Boxplots for both bias correction methods (DBC and SBC) are constructed from the distribution of relative errors from all catchments within each size class (small, medium, and large).**



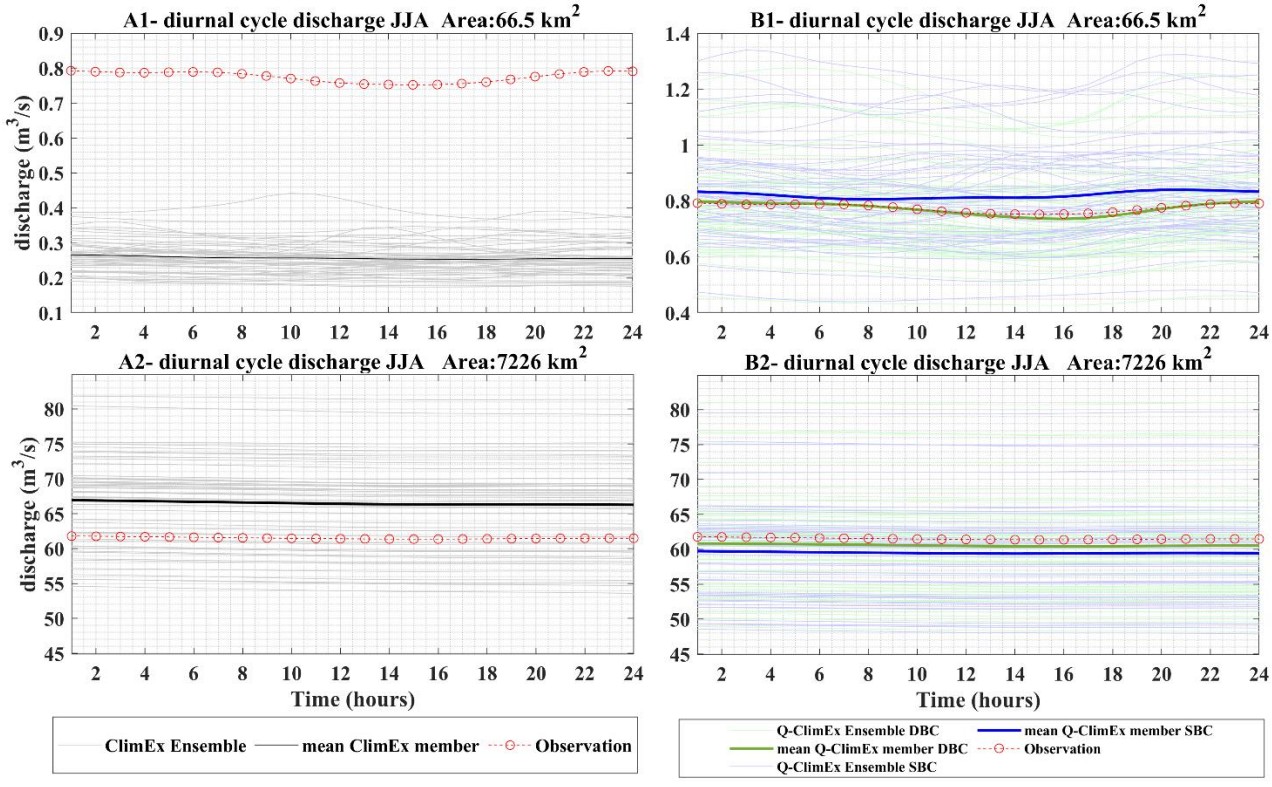

**Figure 10: Annual diurnal cycle of discharge in JJA (Jun, July, August) before bias correction (first column: A1 and A2) and after bias correction (second column: B1 and B2) for two selected catchments. First row is for catchment 02143040 (small size classification) and second row is for catchment 02156500 (large size classification). The observations are shown in red. Streamflow**
**simulations using uncorrected ClimEx members are shown in light grey, and the ensemble mean is in black. Simulations using bias corrected data are in light blue (SBC) and light green (DBC) with the corresponding dark colours showing the ensemble mean. Time is local with 24h corresponding to midnight.**


| | Number of Catchments | Area (km²) | | | Annual Temperature(°C) | | | Annual Precipitation(mm) | | |
|---|---|---|---|---|---|---|---|---|---|---|
| | | Min | Median | Max | Min | Median | Max | Min | Median | Max |
| Small Area | 12 | 66.5 | 268.8 | 468.7 | 9.4 | 11.9 | 15.3 | 967.2 | 1247.3 | 1891.5 |
| Medium Area | 25 | 530.9 | 758.6 | 994.5 | 7.4 | 10.9 | 17.8 | 861.9 | 1072.0 | 2007.8 |
| Large Area | 96 | 1002.3 | 3595.1 | 9885.9 | 7.4 | 11.5 | 19.1 | 804.2 | 1049.6 | 1657.3 |

**Table 1: General Characteristic of the three-catchment size groups**

**Appendix :**

| Catchment ID | | | | | | | | |
|---|---|---|---|---|---|---|---|---|
| 01197500 | 03175500 | 02138500 | 01567000 | 02126000 | 02472000 | 03324300 | 03524000 | 05440000 |
| 01518000 | 03238500 | 02143000 | 01574000 | 02135000 | 02478500 | 03326500 | 03528000 | 05447500 |
| 01520000 | 03303000 | 02143040 | 01628500 | 02156500 | 02479300 | 03328500 | 03540500 | 05454500 |
| 01541000 | 03346000 | 02143500 | 01631000 | 02202500 | 02482000 | 03331500 | 04100500 | 05515500 |
| 01556000 | 03438000 | 03111500 | 01643000 | 02217500 | 02486000 | 03339500 | 04113000 | 05517500 |
| 01558000 | 03443000 | 03361650 | 01664000 | 02228000 | 03011020 | 03345500 | 04115000 | 05518000 |
| 02018000 | 03473000 | 03504000 | 01667500 | 02329000 | 03109500 | 03349000 | 04164000 | 05520500 |
| 02058400 | 03531500 | 03550000 | 01668000 | 02339500 | 03164000 | 03361500 | 04176500 | 05526000 |
| 02118000 | 04201500 | 07261000 | 01674500 | 02347500 | 03168000 | 03362500 | 04178000 | 05552500 |
| 02475500 | 04221000 | 01371500 | 02016000 | 02365500 | 03237500 | 03364000 | 04185000 | 05554500 |
| 03079000 | 05517000 | 01543500 | 02055000 | 02375500 | 03266000 | 03365500 | 04191500 | 05555300 |
| 03161000 | 01372500 | 01548500 | 02083500 | 02383500 | 03269500 | 03451500 | 04198000 | 05569500 |
| 03167000 | 01445500 | 01559000 | 02102000 | 02387500 | 03274000 | 03455000 | 05430500 | 05582000 |
| 03173000 | 01560000 | 01562000 | 02116500 | 02448000 | 03289500 | 03465500 | 05435500 | 05584500 |
| 05592500 | 05593000 | 05594000 | 07029500 | 07056000 | 07290000 | 07363500 | | |

**Appendix 1: USGS ID of the selected MOPEX catchments.**