# Peer review of "Impact of correcting sub-daily climate model biases for hydrological studies"

_Hydrology and Earth System Sciences, 2021_

## Referee Comment (RC1)

**Review of Faghih et al**

**General comments**

Faghih and coauthors investigate the impact of bias-correcting climate models at the sub-daily time scale on streamflow predictions. They find small, but consistent improvement, especially for small catchments. I find the manuscript logically organized and well written although I think several errors are made in English, but I will not go in detail as I am not a native speaker myself either and professional correction seems more appropriate. The goals of the study are clear and so are the conclusions. I have no major comments on the content, but do have some comments on the presentation material and the way this research is embedded in the scientific literature. Overall, I think this manuscript can be accepted subject to minor revisions after the following issues are addressed:

- Self-citations and citing other work. I noted that the manuscript contains 15 self-citations of the second author François Brissette. I strongly wonder whether such a high number of self-citations is truly justified and whether it cannot be reduced. Moreover, other seemingly relevant works are overlooked. For example, but not limited to: Bárdossy and Pegram (2011), Li et al. (2016).
- Figures. The figures and their captions need improvement. Labels 'a','b', etc. are missing; often unclear whether 1 particular year is considered or an average over 24 years; x-axis time in hours has no reference to what is 0, whether this is local time or UTC, mismatches with the text that discusses AM and PM, and over a year and indication of months would still be more logical; for sample watersheds it is sometimes specified which ones they are and sometimes it simply isn't making reproduction impossible; legends often refer to multiple panels, thus placing them outside a panel makes more sense; some figures present 'envelopes' without defining what exactly these envelopes mean in the caption.
- Code and data availability. The code and data availability statement does not contain any information about the availability of the actual code that was generated to produce the results in this paper.

**Specific comments**

L22-23: "Results show small but systematic improvements of streamflow simulations when bias-correcting the diurnal cycle of precipitation and temperature."
Please quantify in a summarized way.

L238: "(variability around the ensemble mean expressed in %)"
since no quantification is actually given, it seems completely irrelevant to note that it can be expressed as a percentage, whereas the statement would remain valid if expressed as a fraction.

L385-386: "It is also well-known that the NSE criterion that was chosen for the hydrological model calibration is more sensitive to high-flows."
Provide a reference for this statement.

L390-391: "A single climate model was used and our results should be replicated with other climate models."
Why? Is there any reason to expect a different result?

L396: "However the MBCn (Cannon, 2018) is arguably the best quantile mapping method available.
Repetition, please delete.

L447: "correcting the diurnal cycle results in better streamflow simulation,"
Please explain 'better' and quantify in a summarized way.

**Technical corrections**

L25: "of summer streamflow on small catchments"
on --> in

L139: "MBCn was chosen …"
No need to start a new paragraph

L162 and L165. The brackets for these references are not correctly placed.

L169: "PET"
Please use the more scientific notation of single symbols, thus $E_p$ instead. See:
https://iahs.info/Publications-News/Other-publications/Guidelines-for-the-use-of-units-symbols-and-equations-in-hydrology.do

L240: "very efficient"
Delete very

L331. Do not write single sentence paragraphs

**References**

Bárdossy, A. and Pegram, G.: Downscaling precipitation using regional climate models and circulation patterns toward hydrology, Water Resour. Res., 47(4), 1–18, doi:10.1029/2010WR009689, 2011.

Li, J., Johnson, F., Evans, J. and Sharma, A.: A comparison of methods to estimate future sub-daily design rainfall, Adv. Water Resour., 110, 215–227, doi:https://doi.org/10.1016/j.advwatres.2017.10.020, 2017.

---

## Author Comment (AC8)

**Response to Reviewers**

We would like to thank the editors and reviewers for their insightful and constructive comments. Because of these comments, the revised version of the manuscript is significantly improved compared to the original one. We have addressed all concerns in this document and most of them in the revised manuscript. Below, we provide a detailed response to each of the reviewers' comments. For convenience, we put the **reviewer comments in black font**, and author responses in blue. In our responses, the line numbers, when specified, refer to the track-change version of the revised manuscript.

**Anonymous Referee #1**

Review of Faghih et al General comments Faghih and co-authors investigate the impact of bias-correcting climate models at the sub-daily time scale on streamflow predictions. They find small, but consistent improvement, especially for small catchments. I find the manuscript logically organized and well written although I think several errors are made in English, but I will not go in detail as I am not a native speaker myself either and professional correction seems more appropriate. The goals of the study are clear and so are the conclusions. I have no major comments on the content, but do have some comments on the presentation material and the way this research is embedded in the scientific literature. Overall, I think this manuscript can be accepted subject to minor revisions after the following issues are addressed:

- Self-citations and citing other work. I noted that the manuscript contains 15 self-citations of the second author François Brissette. I strongly wonder whether such a high number of self-citations is truly justified and whether it cannot be reduced. Moreover, other seemingly relevant works are overlooked. For example, but not limited to: Bárdossy and Pegram (2011), Li et al. (2016).

The second author has been working in the field of *climate change impact assessment* on water resources for the past 20 years.   A lot of work made in our lab in the past is therefore directly relevant to this paper, some of which has been cited by numerous other scientists.   We will plead guilty to being

a bit lazy in finding other relevant work to substantiate some of the claims made in this paper, but not to using irrelevant self-citations. We have added a number of additional relevant citations in the revised version (including the above two suggested references). The following reference have all been specifically added to answer this point. Additional references have also been added in light of other reviewer comments. As a result, this revised version of the paper cites around 100 papers.

Huang, L., Wang, L., Zhang, Y., Xing, L., Hao, Q., Xiao, Y., & Zhu, H. (2018). Identification of groundwater pollution sources by a SCE-UA algorithm-based simulation/optimization model. *Water*, *10*(2), 193.

Muttil, N., & Jayawardena, A. W. (2008). Shuffled complex evolution model calibrating algorithm: Enhancing its robustness and efficiency. *Hydrological Processes: An International Journal*, *22*(23), 4628-4638.

Bárdossy, A. and Pegram, G.: Downscaling precipitation using regional climate models and circulation patterns toward hydrology, Water Resour. Res., 47(4), 1–18, doi:10.1029/2010WR009689, 2011.

Li, J., Johnson, F., Evans, J. and Sharma, A.: A comparison of methods to estimate future sub-daily design rainfall, Adv. Water Resour., 110, 215–227, doi:https://doi.org/10.1016/j.advwatres.2017.10.020, 2017.

Teutschbein, C., & Seibert, J. (2013). Is bias correction of regional climate model (RCM) simulations possible for non-stationary conditions?. Hydrology and Earth System Sciences, 17(12), 5061-5077.

Maraun, D. (2012). Nonstationarities of regional climate model biases in European seasonal mean temperature and precipitation sums. Geophysical Research Letters, 39(6).

Wang, C., Zhang, L., Lee, S. K., Wu, L., & Mechoso, C. R. (2014). A global perspective on CMIP5 climate model biases. Nature Climate Change, 4(3), 201-205.

Ashfaq, M., Bowling, L. C., Cherkauer, K., Pal, J. S., & Diffenbaugh, N. S. (2010). Influence of climate model biases and daily-scale temperature and precipitation events on hydrological impacts assessment: A case study of the United States. Journal of Geophysical Research: Atmospheres, 115(D14).

Ajaaj, A. A., Mishra, A. K., & Khan, A. A. (2016). Comparison of BIAS correction techniques for GPCC rainfall data in semi-arid climate. Stochastic environmental research and risk assessment, 30(6), 1659-1675.

Su, T., Chen, J., Cannon, A. J., Xie, P., & Guo, Q. (2020). Multi-site bias correction of climate model outputs for hydro-meteorological impact studies: An application over a watershed in China. *Hydrological Processes, 34*(11), 2575-2598.

Cannon, A. J., Piani, C., & Sippel, S. (2020). Bias correction of climate model output for impact models. In Climate Extremes and Their Implications for Impact and Risk Assessment (pp. 77-104). Elsevier.

Ayar, P. V., Vrac, M., & Mailhot, A. (2021). Ensemble bias correction of climate simulations: preserving internal variability. Scientific Reports, 11(1), 1-9.

• Figures. The figures and their captions need improvement. Labels 'a','b', etc. are missing; often unclear whether 1 particular year is considered or an average over 24 years; x-axis time in hours has no reference to what is 0, whether this is local time or UTC, mismatches with the text that discusses AM and PM, and over a year and indication of months would still be more logical; for sample watersheds it is sometimes specified which ones they are and sometimes it simply isn't making reproduction impossible; legends often refer to multiple panels, thus placing them outside a panel makes more sense; some figures present 'envelopes' without defining what exactly these envelopes mean in the caption.

We thank you for the relevant comments about the Figures. We have modified several Figures as well as Figure captions. The new Figures as well as brief comments on changes are presented below.

Additional details were added to Figure 3 and 4 including captions. The locations of legends were changed and labels (A, B) were added to the subplots. Times are all local. With 24h corresponding to midnight. Clarifications have been added in the text and AM/PM time have all been modified to a 1-24 hour format.

[Figure]

Figure 3 Annual diurnal cycle of temperature before bias correction (first column: A1 to A4) and after bias correction (second column: B1 to B4) for catchment 02143040. Each row corresponds to a different season: DJF (December, January, February), MAM (March, April, May), JJA (Jun, July, August), SON (September, October, November). The right hand side shows both bias correction methods: Standard Bias Correction (SBC) and Diurnal Bias Correction (DBC). The observations (ERA5) are shown in red. Raw (uncorrected) Climex data is in grey, SBC is in blue and DBC is in green. The envelope defined by all 50 Climex members are shown in the corresponding light colours, whereas the dark coloured lines display the ensemble mean. Time is local with 24h corresponding to midnight.

[Figure]

Figure 4 Annual diurnal cycle of precipitation before bias correction (first column: A1 to A4) and after bias correction (second column: B1 to B4) for catchment 02143040. Each row corresponds to a different season: DJF (December, January, February), MAM (March, April, May), JJA (Jun, July, August), SON (September, October, November). The right-hand side shows both bias correction methods: Standard Bias Correction (SBC) and Diurnal Bias Correction (DBC). The observations (ERA5) are shown in red. Raw (uncorrected) Climex data is in grey, SBC is in blue and DBC is in green. The envelope defined by all 50 Climex members are shown in the corresponding light colours, whereas the dark coloured lines display the ensemble mean. Time is local with 24h corresponding to midnight.

In Figure 5, the transparency of the blue color was modified to a lighter tone which better shows the green color. The caption was rewritten and the location of the legend was changed. Labels were added to each subplot.

[Figure]

Figure 5 Hydrograph annual cycles for four selected catchments.  Catchments A and B are classified as large and medium size respectively. Catchments C and D are classified as small. 0 represents January first at 0h00, and 8760 is December 31st at 24h00.

Labels were added to figure 8 and the caption was modified.

[Figure]

Figure 8 Distribution of the relative error (model-obs)/obs x 100% corresponding to flow quantiles Q5 (A). Q10 (B), Q95(C) and Q99(D). Boxplots for both bias correction methods (DBC and SBC) are constructed from the distribution of relative errors from all catchments within each size class (small, medium, and large)

The location of the legend was changed in Figure 10 and the caption was modified.

[Figure]

Figure 10 Annual diurnal cycle of discharge in JJA (Jun, July, August) before bias correction (first column: A1 and A2) and after bias correction (second column: B1 and B2) for two selected catchments. First row is for catchment 02143040 (small size classification) and second row is for catchment 02156500 (large size classification). The observations are shown in red. Streamflow simulations using uncorrected ClimEx members are shown in light grey, and the ensemble mean is in black. . Simulations using bias corrected data are in light blue (SBC) and light green (DBC) with the corresponding dark colours showing the ensemble mean. Time is local with 24h corresponding to midnight.

• Code and data availability. The code and data availability statement does not contain any information about the availability of the actual code that was generated to produce the results in this paper.

We have modified the *code and availability* section to include the links to all programs and databases used in this study. We have added the list of catchments selected from the MOPEX database as an Appendix (see Table below). Anyone wishing to reproduce this work has now direct access to all data and programs. We don't have an actual 'plug and play' code that does everything in one click. However, the steps are straightforward – data extraction (MOPEX, Climex database, ERA5), hydrological model calibration (GR4J code, SCE-UA code), bias correction (MBCn code), run hydrological model.

Appendix 1 USGS ID of the selected MOPEX catchments.

| Catchment ID | | | | | | | | |
|---|---|---|---|---|---|---|---|---|
| 01197500 | 03175500 | 02138500 | 01567000 | 02126000 | 02472000 | 03324300 | 03524000 | 05440000 |
| 01518000 | 03238500 | 02143000 | 01574000 | 02135000 | 02478500 | 03326500 | 03528000 | 05447500 |
| 01520000 | 03303000 | 02143040 | 01628500 | 02156500 | 02479300 | 03328500 | 03540500 | 05454500 |
| 01541000 | 03346000 | 02143500 | 01631000 | 02202500 | 02482000 | 03331500 | 04100500 | 05515500 |
| 01556000 | 03438000 | 03111500 | 01643000 | 02217500 | 02486000 | 03339500 | 04113000 | 05517500 |
| 01558000 | 03443000 | 03361650 | 01664000 | 02228000 | 03011020 | 03345500 | 04115000 | 05518000 |
| 02018000 | 03473000 | 03504000 | 01667500 | 02329000 | 03109500 | 03349000 | 04164000 | 05520500 |
| 02058400 | 03531500 | 03550000 | 01668000 | 02339500 | 03164000 | 03361500 | 04176500 | 05526000 |
| 02118000 | 04201500 | 07261000 | 01674500 | 02347500 | 03168000 | 03362500 | 04178000 | 05552500 |
| 02475500 | 04221000 | 01371500 | 02016000 | 02365500 | 03237500 | 03364000 | 04185000 | 05554500 |
| 03079000 | 05517000 | 01543500 | 02055000 | 02375500 | 03266000 | 03365500 | 04191500 | 05555300 |
| 03161000 | 01372500 | 01548500 | 02083500 | 02383500 | 03269500 | 03451500 | 04198000 | 05569500 |
| 03167000 | 01445500 | 01559000 | 02102000 | 02387500 | 03274000 | 03455000 | 05430500 | 05582000 |
| 03173000 | 01560000 | 01562000 | 02116500 | 02448000 | 03289500 | 03465500 | 05435500 | 05584500 |
| 05592500 | 05593000 | 05594000 | 07029500 | 07056000 | 07290000 | 07363500 | | |

**Specific comments**

L22-23: "Results show small but systematic improvements of streamflow simulations when bias correcting the diurnal cycle of precipitation and temperature."
Please quantify in a summarized way.

The original sentence:

*Results show small but systematic improvements of streamflow simulations when bias correcting the diurnal cycle of precipitation and temperature.*

has been replaced with:

*Results show relatively small (3 to 5%) but systematic decreases in the relative error of most simulated flow quantiles when bias-correcting the diurnal cycle of precipitation and temperature.*

L238: "(variability around the ensemble mean expressed in %)" since no quantification is actually given, it seems completely irrelevant to note that it can be expressed as a percentage, whereas the statement would remain valid if expressed as a fraction.

Correct. We have removed the latter half of the sentence. The new sentence reads as follows:
*'The relative internal variability (around the ensemble mean) remains the same before and after correction'*

L385-386: "It is also well-known that the NSE criterion that was chosen for the hydrological model calibration is more sensitive to high-flows." Provide a reference for this statement.

Since the NSE criterion is a normalized root mean square error criteria, it naturally follows that it's more sensitive to errors in large values. The automatic calibration algorithms will therefore be more influenced by solutions targeting high flows. We have added the following references to support this.

Krause, P., Boyle, D. P., & Bäse, F. (2005). Comparison of different efficiency criteria for hydrological model assessment. Advances in geosciences, 5, 89-97.

Muleta, M. K. (2012). Model performance sensitivity to objective function during automated calibrations. Journal of hydrologic engineering, 17(6), 756-767.

L390-391: "A single climate model was used and our results should be replicated with other climate models." Why? Is there any reason to expect a different result?

That's a good question. When it comes to the efficiency of bias correction of precipitation and temperature, the answer is a definite no. Quantile mapping approaches are powerful tools to match one distribution onto another. As was shown in the literature, you could bias correct an atmospheric pressure field onto precipitation and perfectly match the target distribution. However, no bias correction method can correct all statistics and particularly so when it comes to joint distribution properties (between P and T in this case). Hydrological models are good spatial integrators of such data, but they are sensitive non-linear integrators. As such, small changes between two climate models (e.g. spatial resolution, interannual variability) could ultimately results in differing streamflow simulations. While we do not expect dramatically different results using other climate models we might see a different sensitivity to catchment size for example. We have modified the text to better reflect the above in the revised version.

L396: "However the MBCn (Cannon, 2018) is arguably the best quantile mapping method available. Repetition, please delete.

Done.

L447: "correcting the diurnal cycle results in better streamflow simulation," Please explain 'better' and quantify in a summarized way.

Has been modified with:

*"Results indicate that correcting the diurnal cycle results in better streamflow simulation, especially for smaller catchments, which have a definite sub-daily response time. For the small catchments, the relative error between observed and simulated flow quantiles was reduced. For example, the median reduction was 5% for the 95th and 99th quantiles, and 4% for the median value of the 20-year flood across all small catchments."*

Technical corrections

L25: "of summer streamflow on small catchments" on --> in

Corrected

L139: "MBCn was chosen ..." No need to start a new paragraph

It was modified accordingly.

L162 and L165. The brackets for these references are not correctly placed.

Corrected.

L169: "PET" Please use the more scientific notation of single symbols, thus Ep instead. See: https://iahs.info/Publications-News/Other-publications/Guidelines-for-the-use-of-units-        symbols-andequations-in-hydrology.do

We have changed PET to Ep throughout the document.

L240: "very efficient"

Delete very

Deleted

L331. Do not write single sentence paragraphs

We reviewed the text to eliminate this and other instances of single sentence paragraphs.

References

Bárdossy, A. and Pegram, G.: Downscaling precipitation using regional climate models and circulation patterns toward hydrology, Water Resour. Res., 47(4), 1–18, doi:10.1029/2010WR009689, 2011.

Li, J., Johnson, F., Evans, J. and Sharma, A.: A comparison of methods to estimate future sub-daily design rainfall, Adv. Water Resour., 110, 215–227, doi:https://doi.org/10.1016/j.advwatres.2017.10.020, 2017.

As discussed above, the above two references (and many others) have been added to the manuscript.

**Comment on hess-2021-236**

Anonymous Referee #2

Review of "Impact of correcting sub-daily climate model biases for hydrological studies", Faghih al.
In this paper the authors present a study on bias correction of precipitation and temperature projections at the sub-daily scale, noting that bias correction of these variables is a necessary ingredient to reliable projections of the hydrological response of catchments to climate change. Overall the paper is well structured and written and the topic is relevant and interesting to the readership of this journal.
Thanks for the kind and constructive comments.

**General comments:**

While the research that is developed is as said well developed and interesting, I was perhaps a little disappointed in the discussion of the results, which although they highlight to some extent the findings, their could have been a little more depth in the exploration of the results, and how these relate to the particular hydro-meteorological processes in the set of catchments considered. The MOPEX dataset is a well-established dataset, and has been studied widely, which also means there is a lot of research to benefit from to support a more in-depth discussion. The main explanatory variable that the authors propose is catchment size, but I could imagine there may well be other explanatory variables. The catchments selected are in a relatively concentrated geographic area, but I would also expect that there may be quite different dominating hydrological processes. I would for example assume that some of the catchments around the Southern great lakes are dominated by spring snowmelt and perhaps convective storms in the summer, while the more coastal catchments east of the Appalachians may well be dominated by frontal precipitation from mid-latitude storm depressions. I myself am not an expert in the climate nor the hydrology of the Eastern USA, but I could imagine that the relative importance of these different hydro-meteorological processes is relevant to the bias correction of precipitation and/or temperature.
We have made several changes to the manuscript following your comments. In particular, several elements have been added to the discussion.  Most of the changes suggested above are discussed in more detail in our answer to the detailed comments below.

It is also clear in the diurnal precipitation patterns, that JJA rainfall is largely convective and thus falls in the afternoon/evening, while in the remainder of the seasons it is perhaps more frontal in nature. How the different types of rainfall that are dominant in a catchment and how this changes may also be important to biases and their correction. Resolution of convection is raised in the introduction, but not discussed much further.

Over the region that is covered in this work, you are correct that a significant part of the JJA rainfall is indeed convective.  With respect to bias correction, there are two issues at play: the ability of the climate model at accurately representing convective precipitation (and other physical processes), and secondly the ability of the bias correction method to correct for those deficiencies. Modern bias corrections methods are extremely powerful and can map any distribution onto another.  For a pure *model output statistic* point of view, it does not matter whether or not the precipitation is convective, orographic or synoptic scale, if it is biased compared to another distribution, it can be corrected.  It is therefore quite easy for example to bias correct Tokyo precipitation data to perfectly match London's precipitation. The question is whether or not it should be done. This is more difficult to answer as it relates to the origin of the biases.  If the biases come from bad model physics it is harder to justify bias correction than if they come from an insufficient representation of the topography. We chose not to address these issues in the original manuscript since while they are very relevant to climate change impact studies, they are a lot less relevant to the very narrow problem we are addressing in this paper, which is simply whether or not sub-daily bias correction is relevant.

We have added additional comments below (first detailed comment L.52) and have decided to modify the discussion to tackle the issue of scale and model physics in the revised version.

I think the discussion could be much strengthened if the author would go beyond catchment size as the explanatory variable. Perhaps it is the case that it is indeed the most significant influence in explaining how important the bias correction is, as well as the bias correction of either variable, but it is worth exploring, and the interest as well as the scientific contribution of the paper would be benefitted by at least some discussion of the different hydrological characteristics of the selected basins. I am sure that there is sufficient other research on the characteristics of the MOPEX dataset to support such a discussion. I would recommend the authors revise the discussion section to provide more depth and insight into the proposed bias correction methods through the lens of the hydroclimatology of the

catchments. I feel some findings are not well addresses (see detailed comments) due to this lack of depth from this perspective.

The main points above are discussed in more details below. As a result, we have significantly modified the discussion in the revised version of the paper to incorporate these elements.

Detailed comments:

Line 52: The authors comment that many GCM and ESM have coarse resolutions of up to 100km (order 1 degree). I would think that it would be better to refer to the resolution in degrees (as this is common practice). Also, this is a fast-moving field, and the resolution of such models has decreased gradually and continues to do so (see e.g. discussion on the convergence of GCM and ESM by Bierkens, WRR (https://doi.org/10.1002/2015WR017173). Also, the HTESSEL model that underlies the ERA5 data that the authors use could be considered a coupled ESM/Land Surface Model (LSM), and has a higher resolution. It may be good qualify the comment here, but also to include this in the discussion. Will this gradual decrease in resolution in space and time reduce the need for such bias correction as proposed here?

Model resolution has been modified to degrees in the revised version. GCM/ESM spatial resolution is indeed decreasing, although this is not consistent across all modelling centers. Many modeling centers have kept the same resolution in the CMIP5 and CMIP6 experiment, opting instead for additional complexity in the models (e.g. carbon cycle, surface model). Since computing time is a function of spatial resolution to the fourth power (x,y,z,t), there are physical limits as to how rapidly model resolution can decrease. Regional climate models have comparatively seen larger decreases in spatial resolution, and especially so for convection-permitting models, which need a horizontal spatial resolution below 2-4km to numerically resolve convection. However, the same computational physical limits still apply to regional climate models, and this increase in spatial resolution has been at the expense of a progressively smaller computational domain. Does a decrease in space and time resolution reduce the need for bias correction? For most impact studies, sadly, our answer to that question is 'no'. Model improvements have been shown to reduce the biases. These improvements come from the increased resolution (better representation of local topography and land surface) and from better physics. However climate models remain an imperfect representation of the real climate system, and the sensitivity of impact models to input data (e.g. precipitation, temperature) will still require some level of post-processing to insure realistic outputs from impact models. The Climex ensemble used in this study has

a very high resolution (0.11º) and still, it clearly requires bias correction. Using uncorrected climate model data results in unrealistic streamflow simulations (e.g. Figure 5). However, with better and higher-resolution models, there is hope that post-processing methods will only end up correcting minor model deficiencies, and not correcting bad physics over a given area (e.g. incorrect precipitation annual cycle). Bias-correction methods such as the one used in this paper are powerful and can map any distribution (good or bad) onto another. However, just because we can, does not mean we should. In a perfect world, every climate model should be evaluated over a given region prior to post-processing its outputs, as it is hard to justify correcting a model with bad physics over a region. In the real world, this is not an easy task, and impact modelers are generally not well equipped for this task and typically end up using all available climate models instead of a careful selection of the best-performing ones. To sum it up, we don't think that better climate models will eliminate the need for bias correction, but rather that it will reduce instances of bad usage of bias correction.

We have substantially modified the discussion to include some of those elements. We have added additional references (including the one suggested above).

Line 110: The figure presenting the different catchments could be bit more informative. The size of the circle as well as the colour are now used to indicate how the catchments fall in the different categories. One of these dimensions is redundant. As there are some larger catchments, the division across the three scales the paper focuses on is not clear. One could perhaps use a different symbol (square, circle, triangle) for the category and colour for the size would be more informative. It is also clear from the table that the sample of small catchments is in fact quite small. How well are these spread over the different hydro-climatologies, or are they concentrated in the more mountainous Appalachians (it is not so clear in the figure)?

The figure was modified accordingly. In the new figure the square, circles, and triangles represent small, medium and large watersheds respectively. As can be seen from the distribution of the squares, the small catchments are distributed across the study area and are not limited to the Appalachians. We would have preferred having a larger sample of small catchments, but it is a limitation of the MOPEX database.

[Figure]

**Figure 2 Distribution of watersheds across North Eastern America. Circles and triangles symbols correspond to small, medium and large catchments respectively**.

Lines 180-185: The authors present the setup of the calibration the model using SCE-UA and N-S as a performance indicator. I think this is an approach reasonably familiar to the HESS readership, so in some cases the description could be more brief (e.g. one could consider leaving out the equation and simply providing a reference). I am also fine with the fact they do not split the sample, considering that a well calibrated model is important but it is not the focus of the study. However, that does raise the question on how this was organised for the bias correction approach. Was a split sample used? This is not clearly described (in the previous section). On line 215 mention is made that if the bias correction was applied at the hourly time scale, then this would have fitted the observations exactly (for the mean I would assume). This suggest that a split sample is not used, thus raising the question of how well the corrections applied are a generalisation of the biases. This may warrant some discussion.

A split-sample approach was not used in this paper as our main goal was to investigate whether or not sub-daily bias-correction would improve hydrological simulations on small catchments. Moving to a validation period brings an additional variably which is the stationarity of the biases. Our guess is that sub-daily biases are likely not stationary (because of internal variability), and that the advantages of the sub-daily bias correction method would be somewhat reduced when tested over an independent

validation period, as found by Chen et al., (2018) in a comparison study of multivariate vs univariate bias correction methods.  This issue has been added to the discussion in the revised version.

Chen, J., et al. (2018). Impacts of correcting the inter-variable correlation of climate model outputs on hydrological modeling. Journal of Hydrology, 560, 326-341.

Line 262: Figure 5 is not very easy to read. It is mentioned that the light blue curves overlap – but it is very difficult to identify what is what in the figure. This could well be improved. It is also not fully clear of the average biases, as the lower part of the different lines is obscured. Given the information the authors want to convey, would it perhaps be more appropriate to present in the form of a flow duration curve? This may well not show the full variability, but it does provide information on the average annual spread. I would suggest the authors try in any case to make the figure a little clearer.

The transparency of the blue line has been changed in the revised version.   It definitely allows for a better comparison between both curves.

[Figure]

Figure 5 Hydrograph annual cycles for four selected catchments. Catchments A and B are classified as large and medium size respectively. Catchments C and D are classified as small. 0 represents January first at 0h00, and 8760 is December 31st at 24h00.

Line 271-274: I think it is important here to be explicit on the question if the climatic characteristics and the dominant processes that shape the regime (pluvial – nival) of the two catchments compared are indeed similar, thus supporting the claim that the diurnal variability being more apparent in the smaller catchment is indeed attributable to catchment size.

We made sure that both catchments were geographically close and displayed similar climate characteristics. We are therefore confident that the main differences come from the catchment size. This is a good point however, and we have added precisions in the revised manuscript and have modified Figure 6 to show the localization of both catchments (see new Figure 6 below).

Here is a detailed description of the catchment main characteristics. A brief account of the similarities has been given in the modified version.

Large Catchment: Station 03528000 CLINCH RIVER ABOVE TAZEWELL, TN

Drainage area is 1474 sq.mi.
Located in hydrologic unit 06010205 at Latitude: 36:25:30N   Longitude: 083:23:54W
Station is in the state of TN (county code 025) and serviced by the TN office.
Channel slope is 6.59 feet/mile
Stream length is 199.00 miles
Mean basin elevation is 2130.00 feet
   Storage: 0.02%    Lakes: N.A.%       Forests: 53.30%     Glaciers: N.A.%
Soil infiltration index is 3.89 inches
Annual precipitation is 45.00 inches
Precipitation intensity is 3.00 inches/24 hour (expected on the average once each two years)
January minimum temperature is 28.00 degrees Fahrenheit

Daily and longer averages for 69 water years are acceptable (all years except partial years).

Small Catchment: Station 02143040 JACOB FORK AT RAMSEY, N. C

Drainage area is 25.7 sq.mi.
Located in hydrologic unit 03050102 at Latitude: 35:35:26N   Longitude: 081:34:02W
Station is in the state of NC (county code 023) and serviced by the NC office.
Channel slope is 119.70 feet/mile
Stream length is 9.80 miles
Mean basin elevation is 1760.00 feet
Percent of contributing drainage area covered by
   Storage: 0.00%    Lakes: 0.00%       Forests: 90.00%     Glaciers: N.A.%
Soil infiltration index is N.A. inches
Annual precipitation is 49.00 inches
Precipitation intensity is 4.00 inches/24 hour (expected on the average once each two years)
January minimum temperature is 32.00 degrees Fahrenheit

[Figure]

**Figure 6 Hydrographs of two sampled catchments for the month of July (744 hourse=31 days * 24 hourse) (small and large size surface area)**

Line 288: Please check the description of the Figure 7. The red crosses appear to be circles. Also, there is a pink/purple cross that is not explained which I assume is the mean of the distribution.

Thanks for the catch. The text was modified accordingly.

Line 297: Figure 7 shows the relative biases of the small, medium and larger catchments after bias correction. The unbiased model results are not shown. Perhaps this was an obvious choice, but it may be valuable to include these, at least in Fig 7, to understand the relative improvement for each of these scales. Also, in Figures 7,8 & 9 it is clear that the diurnal bias correction always results in a more negative bias than the standard bias correction. This is commented on in the text, but the authors do not provide possible explanation. Is the bias corrected precipitation consistently lower; the temperature and consequent evaporation consistently higher? I would assume this is the case, as if it simply a diurnal

redistribution of moisture, then this may not be so consistently seen in the larger catchments. Please comment.

We have not included the boxplots of uncorrected simulations because the biases are up to two order of magnitude larger than for the corrected ones, as can be seen in the Figure below. While these conclusively demonstrate the need for bias correction, it is simply not possible to outline the difference between both bias correction methods if the uncorrected boxplots are shown on the same Figure. We have instead added a sentence in the revised version explaining why the uncorrected model results are not shown.

[Figure]

The reasons as to why the diurnal bias correction results in less flows (we would not say more negatively biases, since Q99 is positively biased for example) is not entirely clear. Since there is less water when the diurnal cycle is corrected, it implies that evapotranspiration is increased. We have tested this on a few catchments (as shown below for one small catchment), and in all cases, this was verified. What we believe is happening is that the climate model diurnal cycle for temperature is flatter than for the observations (see Figure 3 for example). By correcting it, we get higher afternoon temperatures, leading to an increase in potential evapotranspiration (and evapotranspiration), since we are using a temperature

ETP formulation. We have mentioned this in the revised manuscript although we have not added any additional Figures.

Small Catchment:

Average PET of July-MBC-DCC= 0.1077          Maximum PET of July-MBC-DCC=0.3347

Average PET of July-MBC-NDCC= 0.1018          Maximum PET of July-MBC-NDCC=0.3122

[Figure]

Line 342: I find the discussion on the errors in the tipping bucket somewhat suggestive. I agree that at very small rain rates the time of the tip may not well represent the time of precipitation, but I would hardly think this is a major contribution across all catchments and rainfall intensities in the datasets used here. Is there any research that substantiates this statement?

We agree that this is unlikely to be a major contributor. Still this is a type of error that will average itself out over a longer time period (e.g. daily) and is the therefore more relevant to the shorter time steps. We

have modified this section to do a slightly more detailed analysis of potential problems based on the recent work of Segovia-Cardozo et al., (2021), and in the process, we have deemphasized the importance of the time offset mentioned in the original version.

Segovia-Cardozo, D. A., Rodríguez-Sinobas, L., Díez-Herrero, A., Zubelzu, S., & Canales-Ide, F. (2021). Understanding the Mechanical Biases of Tipping-Bucket Rain Gauges: A Semi-Analytical Calibration Approach. *Water*, *13*(16), 2285.

'*Measuring issues related to the use of tipping bucket rain gauges have been reviewed by Segovia-Cardozo et al. (2021). Those issues are an underestimation of total amounts, and especially so for high intensity rainfall and light drizzle, losses from evaporation and non-linear response to rainfall intensity. In addition, at the sub-daily scale, the above may result in small shifts in the actual recording of small precipitation.'*

Line 371: A discussion is provided on the diurnal cycle due to evapotranspiration in the smaller catchments, with this being better represented after bias correction. I agree that this is may well be detectable in flow records for small catchments. However, when looking at figure 3 it would appear that there is a diurnal cycle in the temperature data. Indeed, the temperature in JJA has such a cycle but a clear positive bias, which one would expect would increase evapotranspiration further. Yet in Figure 10 there is no diurnal cycle at all in the unbiased simulations. This does seem somewhat incongruent. It may be good to explore this a little further. Overall Figure 10 can be improved in clarity also as it is not very clear.

Your comments led us to understand that we were not clear enough in our description of Figure 10. There are diurnal cycles for both temperature and evapotranspiration on all catchments, independent of size. This suggest that there is also a surface runoff diurnal cycle which results from the above factors on all catchments, irrespective of size. However, this surface runoff cycle can only be seen on catchment which have a reaction time smaller than approximately 12 hours so that the catchment outlet can register differences between day and night processes. This is what we were trying to show with Figure 10. We have added to the Figure description in the revised version of the manuscript and made changes to the Figure as demanded by Reviewer 1.

---

## Author Comment (AC14)

[revised manuscript text omitted]

**7 Code and data availability**

| 540 | The                                                                                                                       | MOPEX    | climate       | and         | streamflow     | data base | can       | be do     | wnloaded    | from_     | the fo     | ollowing | link:    |  |
|-----|---------------------------------------------------------------------------------------------------------------------------|----------|---------------|-------------|----------------|------------------|-----------|-----------|-------------|-----------|------------|----------|----------|--|
|     | (https://hydrology.nws.noaa.gov/pub/gcip/mopex/US_Data/) (Duan et al., 2006)                                              |          |               |             |                |                  |           |           |             |           |            |          |          |  |
|     | ERAS                                                                                                                      | 5 data   | are availa    | able on     | the Cop        | ernicus C        | limate    | Change    | Service     | (C3S)     | Climate    | Data     | Store:   |  |
|     | https://cds.climate.copernicus.eu/cdsapp#!/dataset/reanalysis-era5-single-levels?tab=form (Hersbach and Dee, 2016).       |          |               |             |                |                  |           |           |             |           |            |          |          |  |
|     | ClimEx data can be downloaded from: https://www.climex-project.org/en/data-access                                         |          |               |             |                |                  |           |           |             |           |            |          |          |  |
| 545 | The -                                                                                                              | GR4J mod | lel (Perrin e | et al., 200 | 3) and Cema    | Neige snow       | modul     | e (Valéry | et al., 201 | 4) are av | ailable on | the Mat  | lab File |  |
|     | Exchange: https://www.mathworks.com/matlabcentral/fileexchange/61720-gr4j-rainfall-runoff-model-deterministic-and- |          |               |             |                |                  |           |           |             |           |            |          |          |  |
|     | stochastic-methods-with-matlab.                                                                                           |          |               |             |                |                  |           |           |             |           |            |          |          |  |
|     | The                                                                                                                       | SCE      | -UA           | global      | optimiz        | ation            | algorith  | m o       | can         | be        | download   | led      | from:    |  |
|     | https:                                                                                                                    | //www.ma | thworks.com   | n/matlabo   | central/fileex | change/767       | l-shuffle | ed-comple | x-evolutio  | n-sce-ua- | method     |          |          |  |

Field Code Changed

[revised manuscript text omitted]

---

## Author Comment (AC15)

**Impact of correcting sub-daily climate model biases for hydrological studies**

Mina Faghih [1], François Brissette [1], Parham Sabeti [1]

[1] Hydrology, Climate and Climate Change Laboratory, École de technologie supérieure, 1100 Notre-Dame West st., Montreal (Canada) H3C1K3

Correspondence to: Mina Faghih (email: Mina.Faghih.1@ens.etsmtl.ca; Tel.: +15149949710)

**Abstract.** The study of climate change impact on water resources has accelerated worldwide over the past two decades. An important component of such studies is the bias correction step, which accounts for spatiotemporal biases present in climate model outputs over a reference period, and which allows realistic streamflow simulations using future climate scenarios. Most of the literature on bias correction focuses on daily scale climate model temporal resolution. However, a large amount of regional and global climate simulations are becoming increasingly available at the sub-daily time step, and even extend to the hourly scale, with convection-permitting models exploring sub-hourly time resolution. Recent studies have shown that the diurnal cycle of variables simulated by climate models is also biased, which raises issues respecting the necessity (or not) of correcting such biases prior to generating streamflows at the sub-daily time scale. This paper investigates the impact of bias-correcting the diurnal cycle of climate model outputs on the computation of streamflow over 133 small to large North American catchments. A standard hydrological modeling chain was set up using the temperature and precipitation outputs from a high spatial (0.11°) and temporal (1-hour) regional climate model large ensemble (ClimEx-LE). Two bias-corrected time series were generated using a multivariate quantile mapping method, with and without correction of the diurnal cycles of temperature and precipitation. The impact of this correction was evaluated on three small (<500 km$^2$), medium and large (>1000 km$^2$) surface area catchment size classes. Results show relatively small (3 to 5%) but systematic  (4-5% in the relative error of mean and different quantiles of flow including Q95 and Q99 and also flood with 20 years return period)  
[revised manuscript text omitted]